# Signature Kernel Scoring Rule: A Spatio-Temporal Diagnostic for Probabilistic Weather Forecasting

**Archer Dodson**                                                                *archerdodson@gmail.com*
*Department of Statistics | University of Warwick, UK*

**Ritabrata Dutta**                                                    *ritabrata.dutta@warwick.ac.uk*
*Department of Statistics | University of Warwick, UK*

**Reviewed on OpenReview:** *https://openreview.net/forum?id=LOLXpt4E5D*

## Abstract

Modern weather forecasting has increasingly transitioned from numerical weather prediction (NWP) to data-driven machine learning forecasting techniques. While these new models produce probabilistic forecasts to quantify uncertainty, their training and evaluation may remain hindered by conventional scoring rules, primarily MSE, which are designed for single time point predictions and ignore the highly correlated data structures present in weather behaviour. This work introduces the signature kernel scoring rule to the domain of weather forecasting, which reframes weather variables as continuous paths to encode temporal and spatial dependencies through iterated integrals. Validated as strictly proper through the use of path augmentations to guarantee uniqueness, the signature kernel provides a theoretically robust metric for forecast verification and model training. Empirical evaluations through weather scorecards on WeatherBench 2 models demonstrate the signature kernel scoring rule's high discriminative power and unique capacity to capture path-dependent interactions. Following previous demonstration of successful adversarial-free probabilistic training, we train sliding window generative neural networks using a predictive-sequential scoring rule on ERA5 reanalysis weather data. Using a lightweight model, we demonstrate that signature kernel based training outperforms climatology for forecast paths of up to fifteen timesteps.

## 1 Introduction

Weather variables exhibit strong spatial and temporal dependencies and are subject to non-stationary behaviour, with large-scale shifting due to seasonal patterns and climate trends. This variability makes uncertainty quantification essential, which is addressed by probabilistic forecasting. Using an ensemble method, the forecasting model makes a set of predictions, each with slightly perturbed weather model or initial conditions, producing a discrete approximation to the full forecast probability density function. Since 2022, new data-driven machine learning weather prediction approaches (MLWP) boosted by GPU acceleration have seen tremendous development. A variety of data-driven modelling approaches now demonstrate higher resolution, quicker computation, and greater performance on conventionally used forecasting scores than their physics-based NWP counterparts (Pathak et al., 2022; Keisler, 2022; Lam et al., 2023; Chen et al., 2023; Bi et al., 2023; Nguyen et al., 2023; Bouallègue et al., 2024a; Lang et al., 2024a).

In order to verify and assess the quality of a probabilistic model's predictions, scoring rules are introduced, which provide summary statistics for the evaluation of forecasts, often based on a measure of distance or divergence from the observed outcomes. Scoring rules can be composites, which can consider multiple criteria to evaluate model performance not reducible to a distance metric, such as forecast calibration or spread.

However, the area of weather forecasting is noticeably limited in the justification provided for the scoring rules in use. In current literature, very few papers are using scores suggested by meteorological agencies, particularly the European Centre for Medium-Range Weather Forecast (ECMWF), despite many using

their Integrated Forecasting System (IFS) and Artificial Intelligence Forecasting System (AIFS) models as references (ECMWF, 2022). A further effect is that there is no standard in scoring rule usage, with each developed model reporting different performance metrics. As a result, there is a marked need to accurately compare the multitude of new data driven forecasting approaches. Moreover, most existing papers use certain scoring rules because of their simplicity, particularly versions of mean absolute error (MAE) and mean squared error (MSE), which have become the convention as more papers are published (Xu et al., 2024). Furthermore, these scores only evaluate marginal forecast distributions at individual time steps, offering no assessment of whether a model correctly captures the temporal and spatial dependency structure over long forecast horizons. Significantly, many forecasting models are trained deterministically and then manipulated into an ensemble. This choice reflects a trade-off between computation efficiency and performance, particularly as the traditional method for generative probabilistic training is adversarial, which is highly computationally expensive (Scher & Messori, 2021).

In this work, we propose and study the efficacy of a new diagnostic based on the *signature kernel score*, which natively operates over entire forecast paths, as a standardising benchmark for probabilistic forecasting tasks involving spatio-temporal dependency, relevant to the particular problem of weather forecasting. The signature kernel was first developed in 2019 as a similarity measure for paths, grounded in rough path theory, and later optimised for efficient computation in 2021 (Kiraly & Oberhauser, 2019) (Salvi et al., 2021). We build on the theoretical framework of the signature kernel as a score, established in 2023, to consider long time step weather forecasting evaluations (Issa et al., 2023). The application of the signature kernel score to the high-dimensional domain of global weather prediction, where complex correlation structures stand to be taken advantage of by path curvature characteristics, remains a significant and unexplored challenge. Recently, Pacchiardi et al. (2024) proposed a framework of predictive-sequential (prequential) scoring rule to train probabilistic forecasting models, which was adapted for training AIFS in recent works (Lang et al., 2024b). Using the signature kernel score and an extension of the prequential framework via sliding window generative models, we show that we can train generative models to provide excellent multi-step probabilistic forecasting in future global weather prediction scenarios. Our contributions are as follows: **1)** We introduce the signature kernel scoring rule for weather model evaluation and training; **2)** We illustrate diagnostic performance with evaluation on two weather scorecards; **3)** We demonstrate probabilistic training of our scoring rule on ECMWF reanalysis weather data.

Section 2 introduces the necessary background of scoring rules and discusses their use in modern weather forecasting literature. The signature kernel scoring rule is established and shown to meet the theoretical requirements of a strictly proper scoring rule. Section 3 demonstrates the diagnostic utility of the signature kernel score as a multiple time step evaluator with weather scorecards, considering the existing models of the IFS ENS (ensemble), Neural GCM (general circulation model), and FuXi Weather. Section 4 contains the simulation results of training on ERA5 reanalysis data, considering performance via standard metrics and visual behaviour across ensemble members and path lengths, along with a discussion on numerical stability.

All figures, scorecard generation code, results, trained models, and training regimes can be found in the following GitHub repository here. Explanation of evaluation metrics used are in Appendix A. The extended theory behind the signature kernel can be found in Appendix B, C, D and E. Extended objective functions and implementation approaches for evaluation and training used in Section 4 are found in Appendix F. An additional illustrative experiment on signature behaviour is shown in Appendix G. Comments on Weather-Bench data are found in Appendix H. Figures of model architecture and additional training results are found in Appendix I and J. Further discussion of extensions and limitations of the work is found in Appendix K.

## 2 A new diagnostic for spatio-temporal probabilistic forecasting

In this section, we introduce a new diagnostic, which is a strictly proper probabilistic scoring rule based on the signature of a path that accounts for both temporal and spatial dependencies to adequately address the needs of probabilistic weather forecasting. First, we briefly review scoring rules commonly used as diagnostics for probabilistic forecasting. In Section 2.2, we introduce the signature of a path. We discuss the extension to weather data observed on discrete time points in Section 2.3 and common augmentations to signatures of a path suitable for the purpose in Section 2.4. In Section 2.5, we propose the signature kernel scoring rule.

## 2.1 Commonly used Scoring Rules for Diagnostic Purposes

Scoring rules provide summary statistics through a numerical score for the purpose of forecast evaluation. This score is dependent on both the predictive distribution and the true event or value that is observed. The primary function of scoring rules is to measure the quality of probabilistic forecasts, where different rules, using different measures of quality, reward different forecasting behaviours. In the field of climate model predictions and meteorology, this task falls under forecast verification (Gneiting & Raftery, 2007).

Given a scoring rule $S$, and a forecaster's predictive distribution $P \in \mathcal{F}$, the score for a specific event $y \in \Omega$ being observed is $S(P, y)$, where $\mathcal{F}$ is a family of distributions and $\Omega$ is the outcome space. This score is a single value on the real line $\mathbb{R}$. A critical property of well designed scoring rules is that they are strictly proper. Given any prediction distributional forecast $P \in \mathcal{F}$ and the true observation distribution $Q \in \mathcal{F}$, the following holds for a proper scoring rule where better scores are minimised:

$$\mathbb{E}_{Y \sim Q}\left[S(Q, Y)\right] \leq \mathbb{E}_{Y \sim Q}\left[S(P, Y)\right]$$

A strictly proper scoring rule further ensures the equality holds if and only if $P = Q$, i.e. $Q$ is the unique minimiser. The inequality direction can be changed depending on whether the goal is maximisation or minimisation of the score, with minimisation taken to be the objective throughout the rest of this work without further reference. Strict propriety is necessary to prevent attempts at gaming the scoring rule by hedging, i.e. providing predictions that are not truthful, but are produced to achieve better scores. Serious issues potentially arise when a forecaster isn't encouraged to report their true belief, and in the context of truthfully predicting the weather, improper scoring rules are a detriment to model performance. During model training, strictly proper scoring rules naturally lend themselves as loss functions by guaranteeing that the true distribution is the unique expected minimiser of the score.

Table A.1 (Appendix) displays a summary analysis of the scoring rules in use among eight of the leading ML data-driven weather forecasting models between 2022-2024, determined by inclusion in the Google WeatherBench 2 framework (Rasp et al., 2020). Root Mean Squared Error (RMSE) is the most commonly used diagnostic, used on geopotential at 500 hPa and temperature at 850 hPa in every paper reviewed. These variables hold significant meteorological justification for underlying weather systems, and are the primary variables of focus for this paper (ECMWF, 2024a). There are limitations associated with training on RMSE, notably that it results in overly smooth forecast distributions by penalising bold predictions relevant for quantifying extreme weather risk (Bouallègue et al., 2024a). The forecast activity should therefore be a highly significant metric to contextualise differences in RMSE, but is only used in two papers reviewed. As mentioned in WeatherBench, RMSE is chosen primarily because of its ease to compute, and not due to a greater significance in the context of weather prediction (Rasp et al., 2020). Along with undesirable penalising behaviour, there is no spatial or time dependency involved. RMSE measures the average magnitude of differences between predictions and observations, without capturing any correlation or structure within the data. Moreover, despite a majority of models being probabilistic, most evaluation focuses on the accuracy of the ensemble mean point forecast, especially in relation to RMSE. Consequently, forecast uncertainty is under-represented in the analysis.

## 2.2 Signature of a Continuous Path

In order to directly encode spatial and temporal dependencies, the signature kernel score can be proposed as a standardising scoring rule for probabilistic weather prediction models. This scoring rule must first be understood through an investigation into the signature of a path from rough path theory, requiring a reframing of weather data.

A d-dimensional path $X$ is defined as a continuous mapping from an interval $[a, b]$ to $\mathbb{R}^d$, written as $X : [a, b] \to \mathbb{R}^d$ (Chevyrev & Kormilitzin, 2025). We often consider paths parametrised by time, with $X_t = \{X_t^1, X_t^2, ..., X_t^d\}$ representing the position in space through time with $t \in [a, b]$. Each $X^i$ is likewise a path, with $X^i : [a, b] \to \mathbb{R}^1$. A common form of time dependent path is simply defined as $X_t = \{X_t^1, X_t^2\} = \{t, f(t)\}$, which is a generalised time series consisting of successive measures in one dimension over an interval. In this section, we consider smooth paths which are infinitely continuously differentiable, i.e. $C^\infty$ (Chevyrev & Kormilitzin, 2025).

Next, we consider what a path integral represents, where we integrate one path against another. Given paths $X, Y : [a, b] \to \mathbb{R}$ we define the integral:

$$\int_a^b Y_t dX_t = \int_a^b Y_t \frac{dX_t}{dt} dt$$

This forms the basis for how signatures are constructed. For any dimension $i$ of our $d$-dimensional path $X : [a, b] \to \mathbb{R}^d$, we can consider the respective univariate path of $X^i$ and define the single iterated integral as follows:

$$S(X)_{a,b}^i = \int_{a<s<b} dX_s^i = X_b^i - X_a^i = \Delta X_{[a,b]}^i$$

Early iterations are easy to interpret, with the first order integrals being the range of the path in a specific dimension, resulting in $d$ first order terms. We can extend to second order iterations, relating to the encompassed area of the path, where we consider any two dimensions of the path $i_1, i_2 \in \{1, ..., d\}$:

$$S(X)_{a,b}^{i_1,i_2} = \int_{a<s<b} S(X)_{a,s}^{i_1} dX_s^{i_2} = \int_{a<r<s<b} dX_r^{i_1} X_s^{i_2}$$

We extend iteration to the k-iterated integral, for any collection of indexes $i_1, ..., i_k \in \{1, ..., d\}$, we define:

$$S(X)_{a,b}^{i_1,...,i_k} = \int_{a<t_k<b} ... \int_{a<t_1<t_2} dX_{t_1}^{i_1} ... dX_{t_k}^{i_k}$$

These higher order terms no longer have simple geometric interpretations. See appendix B for a further geometric intuition of lower order terms. The signature of path X, denoted by $S(X)_{a,b}$ on the set $[a, b]$ is the infinite collection of all iterated integrals of X, which can be written as an infinite sequence of real numbers:

$$S(X)_{a,b} = (1, S(X)_{a,b}^1, ..., S(X)_{a,b}^d, S(X)_{a,b}^{1,1}, S(X)_{a,b}^{1,2}, ...)$$

The zeroth order term is considered to be 1, by convention. This infinite signature captures all of the sequential dependencies in the path, with each order of term capturing increasingly complex relationships across space over time. The properties of the signature can be compared to moments in statistics. While moments capture characteristics of a distribution, the signature characterises the temporal and structural data within a path. As the moments determine a distribution, albeit under some conditions, the signature likewise uniquely characterises a path, up to translation and time parametrisation. Given the properties of capturing path behaviour, with terms of the signature being good candidates for characteristic features, the full signature contains all the information we want to extract from a path for a machine-learning algorithm to understand the behaviour. See Appendix C for the extension to discrete data.

## 2.3 Signature Augmentations

As previously mentioned, the current framing of the signature uniquely characterises a path up to translation and time parametrisation. Both of these pose issues for use as an evaluation metric and for use in a scoring rule. Learning the behaviour of weather path curvature is not enough if the forecast has significant bias via translation shifts. Weather has clear temporal cycles on the level of days, months, and years. The signature needs to relate path behaviour to time to evaluate a forecast. We introduce the following augmentations (Morrill et al., 2021). The basepoint augmentation $\phi^b : \mathcal{S}(\mathbb{R}^d) \mapsto \mathcal{S}(\mathbb{R}^d)$ is defined as:

$$\phi^b(\mathbf{x}) = (\mathbf{0}, \mathbf{x}_1, ..., \mathbf{x}_n)$$

This augmentation adds an initial point to all paths, which is a zero vector of dimension $d$. The signature is now sensitive to translation, which can be clearly observed from the new diagonal terms. All diagonal terms now solely depend on the magnitude of the endpoint coordinate in each dimension:

$$S(X)_{a,b}^{i,...,i} = \frac{(X_b^i - X_a^i)^k}{k!} = \frac{(X_b^i)^k}{k!}$$

The time augmentation $\phi_t : \mathcal{S}(\mathbb{R}^d) \mapsto \mathcal{S}(\mathbb{R}^{d+1})$ is defined as:

$$\phi_{\mathbf{t}}(\mathbf{x}) = (t_1, \mathbf{x}_1), ..., (t_n, \mathbf{x}_n)$$

This augmentation adds another dimension to the path, which carries the time of the observation at that point. This allows for different or irregular sampling between forecast and observations. This is critical for weather forecasting, as forecasts and observations are regularly predicted and sampled at different intervals. Moreover, the effect of missing values is mitigated, as the time dimension aligns true positions with all existing values correctly. These augmentations combined remove any invariances and guarantee the uniqueness of the signature. This property is critical for the performance of a scoring rule involving the signature kernel, as it is required for strict propriety, discussed in the following subsection. See Appendix D for further discussion of signature augmentations.

## 2.4 The Signature Kernel Scoring Rule

The previous sections introduced the idea, intuition, and practical concerns with the signature of a path. Nonetheless, the signature remains an infinite sequence of real numbers, which still can't be used in its current form. The primary concept involved is the kernel trick, which allows data to be analysed in a higher (or infinitely high) dimensional space, without explicitly computing the mapping, i.e. the full signature. A kernel is a positive semidefinite function that takes vectors in the original spaces as inputs, and returns the inner product of the vectors in the feature space (Wilimitis, 2019). We can define a mapping $k : \mathcal{X} \times \mathcal{X} \to \mathbb{R}$ to be a kernel if there exists a Hilbert space $\mathcal{H}$ (feature space), and a feature map $\phi : \mathcal{X} \to \mathcal{H}$, such that for all $x, z \in \mathcal{X}$ we have:

$$k(x, z) = \langle \phi(x), \phi(z) \rangle_{\mathcal{H}}$$

Equivalent definitions include the symmetric positive-semidefinite Gram matrix. In this case, we would like to consider our forecast and observation paths X and Y to be the vectors in the original space, with the signature of the paths being the feature space. The signature kernel is simply the inner product of two signatures:

$$K^{Sig}(X, Y) = \langle S(X), S(Y) \rangle$$

which maps to a single score. However, to apply the kernel trick, the construction must take place in a reproducing kernel Hilbert space (RKHS). This guarantees the complete inner product structure that is required (Xu et al., 2023). If the original space $\mathcal{Y}$ is not itself a Hilbert space, we must first lift each path in $\mathcal{Y}$ to the RKHS $\mathcal{H}$ induced by a static kernel $k$ on $\mathcal{Y}$ via the canonical feature map. This step also allows us to incorporate a similarity structure on $\mathcal{Y}$. With sequential information, lifting the space before considering the signature would be a good learning strategy regardless of whether $\mathcal{Y}$ is already a Hilbert space (Salvi et al., 2021). Given an RKHS$(\mathcal{H}, k)$ with a kernel $k$ on $\mathcal{Y}$, the reproducing property lets us define a general static kernel as follows:

$$k : \mathcal{Y} \times \mathcal{Y} \to \mathbb{R} \text{ equals } k(u, v) = \langle k_u, k_v \rangle_{\mathcal{H}}$$

Taking the canonical feature map induced by kernel $k$ as $k_u$ or $k_v \in \mathcal{H}$, we can now write the signature kernel as:

$$K^{Sig}(X, Y) = \langle S(k_x), S(k_y) \rangle_{\mathcal{H}}$$

$S(k_x)$ is theoretically guaranteed to capture all of the information about the data X, while observing the path through the kernel k, where the canonical feature map is in the space of paths in $\mathcal{H}$, $k_x \in \mathcal{P}_{\mathcal{H}}$ (Kiraly & Oberhauser, 2019).

In practice, this kernel is the solution of a Goursat partial differential equation (PDE) (Salvi et al., 2021). In fact, the previous condition of piecewise linearity was a necessary condition to be solved by a Goursat PDE. While other methods of solving the kernel exist, involving sequentialised kernels, the time complexity is at least $\mathcal{O}(dl^2)$, for a path with dimension $d$ and length $l$ (Kiraly & Oberhauser, 2019). Provided the number of GPU threads exceeds the size of the discretisation grid chosen for PDE evaluation on (controlled by dyadic order), the time complexity reduces to $\mathcal{O}(dl)$ (Salvi et al., 2021). This breaks the quadratic relationship with path length, making the Goursat PDE approach the most suitable method for implementation, used by the

associated sigkernel Python package. While the constant factor involved in solving the PDE means that evaluation takes longer in practice than point-wise metrics like RMSE or CRPS, the signature kernel scales at the same asymptotic rate. This property ensures it remains feasible for high-dimensional, long-range atmospheric data. See Appendix G for an additional discussion on discriminative power affecting training time. Finally, we can define the signature kernel score considering path data over the time interval $1:t$ and longitude and latitude $i, j$. Denoting the set of ensemble forecast paths as $F_{i,j,1:t}$ and the observation path as $o_{i,j,1:t}$, this score is as follows:

$$\phi(F_{i,j,1:t}, o_{i,j,1:t}) = \mathbb{E}_{X,X' \sim F_{i,j,1:t}} \left[ K^{Sig}(X, X') \right] - 2\mathbb{E}_{X \sim F_{i,j,1:t}} \left[ K^{Sig}(X, o_{i,j,1:t}) \right] \tag{1}$$

Using this score, we are considering both the spread of the forecasted paths in the first term and the closeness, measured by the signature kernel, of the forecasted paths to the true observed paths in the second term. We can note the similarity between this kernel score and the energy score, with the apparent sign switch as a result of kernels measuring similarity versus distance norms representing dissimilarity.

**Theorem 1.** *Under a characteristic static kernel, with paths of bounded variation on a compact domain, and uniqueness assured via augmentations, the signature kernel score described in Equation 1 is strictly proper.*

The strict propriety of the score ensures truthful predictions, as the true path distribution is the unique minimiser of the expected score. Appendix E provides the full proof.

Additionally, we will define the kernel distance as the primary metric to be used on deterministic forecasts. Using the fact that the kernel defines an inner product distance on X and Y within the transformed space $\langle S(k_x), S(k_y) \rangle$, we can instead consider the Euclidean distance on this space (Phillips & Venkatasubramanian, 2011). With a non-ensemble forecast $f_{i,j,1:t}$, the distance is without expectation, with the distance across time $1:t$ and longitude, latitude $i, j$ as:

$$d^2(f_{i,j,1:t}, o_{i,j,1:t}) = K^{Sig}(f_{i,j,1:t}, f_{i,j,1:t}) + K^{Sig}(o_{i,j,1:t}, o_{i,j,1:t}) - 2K^{Sig}(f_{i,j,1:t}, o_{i,j,1:t})$$

We now have a distance measure, as opposed to a kind of similarity measure calculated using the inner product of $\langle S(k_x), S(k_y) \rangle$. Under this measure, all values are non-negative, with the distance equalling zero when observations and forecasts have the same path. Ignoring expectations, the difference between the score and distance is the addition of the term $K^{Sig}(Y, Y)$, which represents the self-similarity of the observation path. As will be shown, this will improve the interpretability of deterministic forecasts by generally making distance monotonic, with a greater focus on forecasting skill than forecast spread.

There is a remaining choice of static kernel to be used in the signature kernel scoring rule. This work will consider the radial basis function (RBF) kernel:

$$K(\mathbf{x}, \mathbf{x}') = \exp \left( \frac{||\mathbf{x} - \mathbf{x}'||^2}{2\sigma^2} \right)$$

for all $\mathbf{x}, \mathbf{x}' \in \mathbb{R}^d$. The RBF kernel changes how the data points in the sequence are related, by introducing a notion of locality, scaled by the hyperparameter $\sigma$. This changes how close values are in kernel space, which results in different behaviours being captured by the signature kernel. The RBF may be better suited to the non-linear relationships present in weather data, but requires greater consideration during implementation to correctly tune $\sigma$. When using the RBF, $\sigma$ was chosen to be equal to 1, following positive experimental results (Issa et al., 2023). Other kernels can be used, such as the more interpretable linear kernel, but it suffers significantly from numerical instability and requires much greater attention to the scale of the input data. Refer to Section 4.3 for further details.

Moreover, dyadic order for the PDE solver needs to be chosen. Higher dyadic orders result in higher accuracy, at the cost of calculation speed. Given our inputs are significantly scaled, such that a maximum value is roughly equal to 1, experimental results show that dyadic orders of 0 or 1 are likely sufficient (Salvi et al., 2021). We implement a dyadic order of 1, following previous work (Issa et al., 2023). The use of the Goursat PDE solver approach also affects what weather variables can be considered. In the setting of piecewise-linear fixed dimensional paths, which move in all dimensions, typically none of the elements of the

signature are zero. This is referred to as a dense signature. On the other hand, rainfall data is generally zero-inflated, particularly for certain ranges of latitude-longitude coordinates. As a result, this may result in a sparse signature, with many terms equal to zero. Methods which handle sparse signatures require significant overhead, which can be avoided if we stay in contexts resulting in dense signatures (Reizenstein & Graham, 2018). Nevertheless, extending this work to explicitly accommodate sparsity remains a valuable direction for future work, as it would enable more robust handling of zero-inflated weather phenomena like precipitation.

## 3 WeatherBench 2.0: Signature Kernel Score in Practice

The signature kernel scoring rule and distance on probabilistic and deterministic forecasts respectively are implemented in a notably different way than standard scores. On WeatherBench 2, scores such as RMSE are calculated at every prediction time lag. Most models provide 30 to 60 values of forecasted data from their initialisation time, spaced by twelve or six hours apart, equivalent to 360 hours or 15 days. The maximal forecast lead time for a model corresponds to when performance is no longer significantly different from predicting the average climatology. These values are averaged across all initialisation times and all locations using latitude weighting, to report score against time lag. The ECMWF then calculates their headline scores by considering the maximal forecast lead time required to reach a certain score (ECMWF, 2022).

The signature kernel score can not be calculated this way. A path is a temporally related sequence, not a collection of values across space at one particular time. The signature kernel score is instead calculated across chronologically sequential paths of length $k$ of the predicted data. Instead of having a value calculated for each prediction time lag, a value is determined for each path length from the initialisation time. Similar to the previous scores, these values can be averaged across all initialisation times. Further details of implementation for model evaluation use can be found in Appendix F.

The following two scorecards are created from a 10% initialisation sample evenly distributed across the year 2020. Scorecard variables are chosen based on the variable overlap between models of the ECMWF headline variables (ECMWF, 2024b). In Figure 1, we compare the FuXi Weather model and IFS ENS Mean across deterministic metrics and signature kernel distance with 60 timesteps of six hours, equivalent to 15 days. In Figure 2, we compare Neural GCM and IFS ENS across probabilistic metrics and signature kernel score with 30 timesteps of twelve hours, equivalent to 15 days. In both cases, IFS is taken as the baseline, with normalised percentage differences in metrics against its comparison being shown in colour. Darker blues signify greater positive performance for the target model as opposed to the baseline, with the opposite true of darker reds. Following ECMWF standards, calculations are split by region, with the Northern Hemisphere containing the latitude slices from 20 to 90 degrees, the Tropics containing -20 to 20, and the Southern Hemisphere containing -90 to -20. There is no overlap between regions.

In Figure 1, the domain for anomaly cross correlation (ACC) and MSE is the 60 future lead times, while the signature kernel scoring rule (SIGK) is displayed against path length between 2 and 59. Despite this difference, there is general agreement across metrics over model performance at different times. However, some notable behavioural differences are detected. The signature kernel distance measure has generally sharper discrimination for positive FuXi performance, seen clearly in the Southern Hemisphere evaluation. The signature kernel also changes model performance evaluations in certain areas, unanticipated by the standard metrics. Northern and Southern Hemisphere 10 meter wind speed improves for FuXi performance for longer path lengths with signature kernel evaluation, differing from worse performance at later time leads for ACC and MSE. This implies model predictions have resulted in more appropriate path behaviours, despite specific values being worse.

In Figure 2, the domain for continuous ranked probability score (CRPS) and ensemble (ENS) MSE is the 30 future lead times, while SIGK is displayed against path length between 2 and 29. In this scorecard, Neural GCM demonstrates superior performance across nearly all metrics and times, compared to the IFS ensemble baseline. The signature kernel score once again appears sharper, but has a notable shift to stronger predictions for later path lengths, as opposed to initial lead times, as seen clearly in Southern Hemisphere variables. Notably, the same model performance changes seen in geopotential in the Northern Hemisphere by CRPS and ENS MSE are also observed by the signature kernel. However, geopotential at 850 hPA in the

Northern Hemisphere does not see a reversal in model performance by the signature kernel. This is due to the path length versus lead time comparison, where the change is small enough on the path length scale to not cause a complete reversal. These behaviours reflect both the unique way the signature kernel score picks up on the structure of the data compared to existing metrics, along with the full path length evaluation approach. See Appendix G for an additional comparative simulation. These differences may support the use of the signature kernel in model evaluation alongside a standard lead time metric.

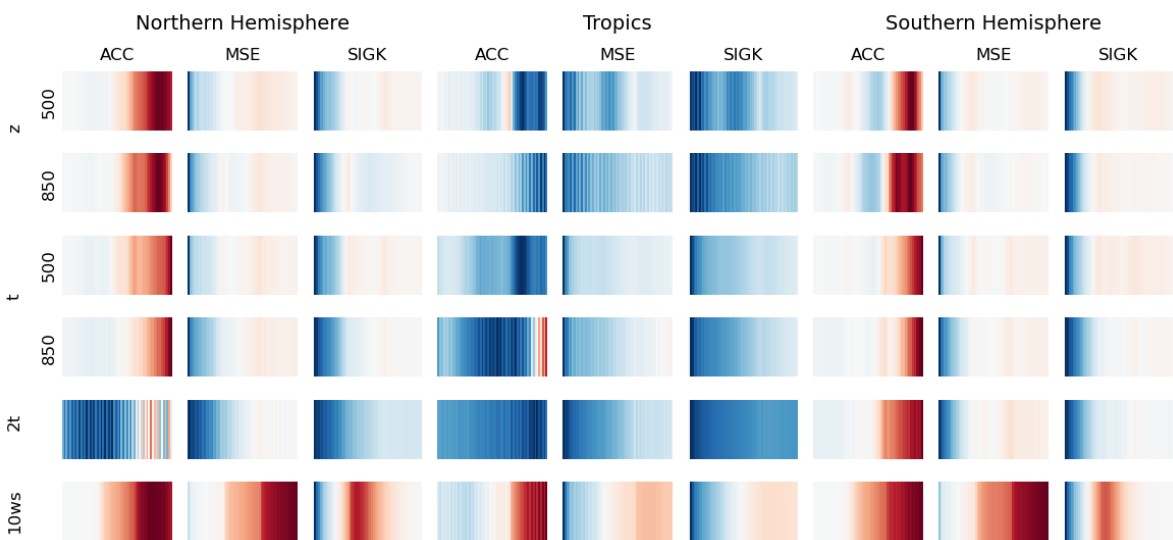

Figure 1: Deterministic scorecard comparison of FuXi and IFS Mean across weather variables and metrics, over three regions of the globe. Darker blues signify positive metric performance for FuXi as compared to the IFS Mean metric value, evaluated over 15 days across geopotential (z) at 500 and 850 hPa, temperature (t) at 500 and 850 hPa, two meter temperature (2t), and 10 meter windspeed (10ws).

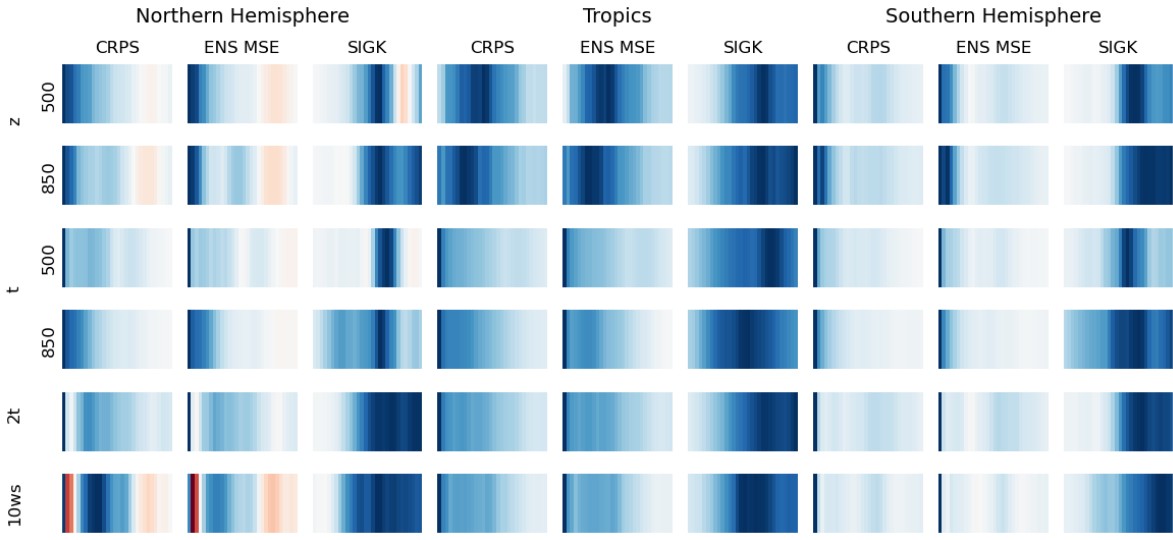

Figure 2: Probabilistic scorecard comparison of Neural GCM and IFS ENS across weather variables and metrics, over three regions of the globe. Darker blues signify positive metric performance for Neural GCM as compared to the IFS ENS metric value, evaluated over 15 days across geopotential (z) at 500 and 850 hPa, temperature (t) at 500 and 850 hPa, two meter temperature (2t), and 10 meter windspeed (10ws).

# 4 Training a Multi-step Probabilistic Forecaster

We employ generative models to provide probabilistic forecasts. These models, specifically generative neural networks, work by transforming randomly sampled latent variables into samples on the required output space. They achieve this by learning a mapping $G$ from a low dimensional simple distribution, $z \sim \mathcal{N}(0, I)$, to our observations $x$, $G : z \to x$. The application of deep neural networks allows our generative model to learn and capture the complex distribution by applying non-linear functions and working with hierarchical representations. This type of model is highly appropriate to the problem of weather forecasting as generative networks predict probabilistic outputs by design, by generating samples as opposed to point predictions. However, the common way to train a generative network, via generative adversarial training, is unstable, suffers from mode collapse, and has a training objective which has not been extended to temporal data (Salimans et al., 2016). As a result, we implement the predictive-sequential (prequential) scoring rule to train generative neural networks to minimise scoring rules, first introduced by Pacchiardi et al. (2024). We first consider the one step ahead prequential scoring rule formulation, before extending to a prequential scoring rule for pathlengths $l$ for sliding window generative models.

## 4.1 Generative Models and the Prequential Scoring Rule

When considering multivariate time series, which is the general case of weather forecasting, we are no longer able to consider observations $y_i$ as originating from a time-independent distribution $P$. Instead, given such a process, $(\mathbf{Y}_1, \mathbf{Y}_2, \ldots, \mathbf{Y}_T)$, with $\mathbf{Y}_t$'s generally not independent, we associate each time RV $\mathbf{Y}_t$ with the marginal distribution $P_t$. Additionally, we can consider the joint distribution $P_{r:s}$ of the sequence $\mathbf{Y}_{r:s}$ and the conditional distribution $P_t(\cdot|\mathbf{y}_{u:v})$ of $\mathbf{Y}_t$ given past realisations $\mathbf{y}_{u:v}$.

The general forecasting problem is that we are given a set of $k$ past realisations, and we want to predict the next observation, $k+1$. In general, we can consider any specific future observation $k+l$, but for simplicity, we consider the immediately sequential step. Having observed $t$ previous values, we employ a generative model, with parameters $\phi$ to give us a forecast distribution conditioned on the previous window of k observations: $P_{t+1}^\phi(\cdot|y_{t-k+1:t})$. We can then evaluate the performance of the forecast at time step $t+1$ with a given score S:

$$S(P_{t+1}^\phi(\cdot|y_{t-k+1:t}), y_{t+1})$$

To evaluate the quality of the overall forecasts, we would compute a summation of the score across all possible times, adjusted by window size:

$$\sum_{t=k}^{T-1} S(P_{t+1}^\phi(\cdot|y_{t-k+1:t}), y_{t+1})$$

Indeed, for such a one step forecast, optimisation over $\phi$ through minimisation of the above term selects for the parameters that yield forecasts whose average score across the training data is optimal. Under conditions of stationarity and mixing for $(\mathbf{Y}_1, \mathbf{Y}_2, \ldots, \mathbf{Y}_T)$, it has been proven that the empirical minimiser $\hat{\phi}$ is consistent, i.e. converges to the minimiser of the expected prequential scoring rule (Pacchiardi et al., 2024). Therefore, to optimise $\phi$ via stochastic gradient descent, we need to obtain unbiased estimates of $\nabla(P_{t+1}^\phi(\cdot|y_{t-k+1:t}), y_{t+1})$, which occur when S is strictly proper and defined by expectation over $P^\phi$. Critically, this is true of the signature kernel scoring rule, seen in terms of expectation above in Equation 1, as is standard for a general kernel score.

We next extend the prequential scoring rule for use in training paths of forecasts. Importantly, for a sliding window generative model we condition only on a previous window of $k$ observations for each step. For a given forecast initialised at time $t$, the entire path must be initialised on the same set of observational data, with later points in the path not affected by further observations. We build a predicted path of length $l$ via recursion on the one step generations, shifting the initial window to exclude the oldest observation and introduce the latest generated samples $P_{t+1}^\phi(\cdot|y_{t-k+1:t}) = \hat{y}_{t+1}$.

We demonstrate this behaviour for a path length of two, defining our path predictions of length $l$ as $P_{t+1:t+l}^{\phi}(\cdot|y_{t-k+1:t})$. Therefore, we can write the full prequential score $S(P_{t+1:t+2}^{\phi}(\cdot|y_{t-k+1:t}), y_{t+1:t+2})$ as:

$$S((P_{t+1}^{\phi}(\cdot|y_{t-k+1:t}), P_{t+2}^{\phi}(\cdot|y_{t-k+2:t}, P_{t+1}^{\phi}(\cdot|y_{t-k+1:t}))), y_{t+1:t+2})$$

Considering each $P_{t+1}^{\phi}(\cdot|y_{t-k+1:t})$ as $\hat{y}_{t+1}$, we can rewrite the equation:

$$S(P_{t+1:t+2}^{\phi}(\cdot|y_{t-k+1:t}), y_{t+1:t+2}) = S((\hat{y}_{t+1}, P_{t+2}^{\phi}(\cdot|y_{t-k+2:t}, \hat{y}_{t+1})), y_{t+1:t+2})$$

Further denoting all $P_{t+l}^{\phi}(\cdot|y_{t-k+l:t}, \hat{y}_{t+1:t+l-1})$ for $l \leq k$ and $P_{t+l}^{\phi}(\cdot|\hat{y}_{t+l-k-1:t+l-1})$ for $l > k$ as $\hat{y}_{t+l}$ given $l \geq 2$, we can write the prequential score for any path length assuming $l > k$ for convenience:

$$S(P_{t+1:t+l}^{\phi}(\cdot|y_{t-k+1:t}), y_{t+1:t+l}) = S((\hat{y}_{t+1}, \ldots, \hat{y}_{t+l-1}, P_{t+l}^{\phi}(\cdot|\hat{y}_{t+l-k-1:t+l-1})), y_{t+1:t+l})$$

The full sliding window generation process is displayed in Figure 3. Recall that each prediction $\hat{y}_l$ is an ensemble, consisting of $e$ ensemble members, where each member requires a latent variable $z$ for each time step. In fact, for learning purposes, the matrix of latent variables $\mathbf{Z}$ is static, resulting in a unique noise term for each combination of ensemble member $e$ and path position $l$, regardless of initialisation time $t$. The result of generation is $e$ paths of length $l$, which can finally be evaluated with a chosen score.

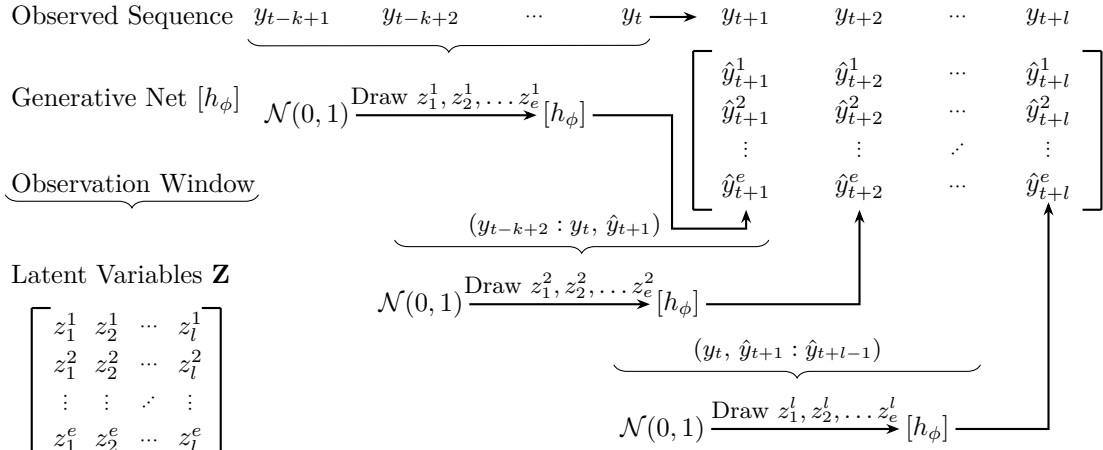

Figure 3: Diagram of the sliding window prequential generation. The standard one-step prequential generation is applied to each observation window, shifting by one into the newly predicted samples each iteration. The generative net receives designated latent variables $z_l^e$, corresponding to ensemble member $e$ and path $l$.

We finally optimise over all possible initialisation times, adjusted by fixed window size $k$ and fixed path length $l$, with our training objective function for a suitable score $S$ as follows:

$$\hat{\phi}_{k,l} = \arg\min_{\phi} \sum_{t=k}^{T-l} S(P_{t+1:t+l}^{\phi}(\cdot|y_{t-k+1:t}), y_{t+1:t+l})$$

The empirical minimiser $\hat{\phi}_{k,l}$ for the sliding window generation converges to the minimiser of the expected prequential scoring rule, under the same conditions as the non-sliding implementation. We produce unbiased estimates of $\nabla_{\phi} S(P_{t+1:t+l}^{\phi}(\cdot|y_{t-k+1:t}), y_{t+1:t+l})$, allowing training to a unique minimum in convergence.

The objective function for the patched signature kernel score, for patches $p \in \mathcal{P}$, initialisation time $t$, path length $l$, and ensemble size $m$, is as follows:

$$\hat{\phi}_{l,m} = \arg\min_{\phi} \sum_{t=k}^{T-l} \left( \sum_{p\in\mathcal{P}} \phi_p(X, y_{t+1:t+l}) \right)$$

where $\phi_p(X, y_{t+1:t+l})$ with $X_p \sim P_{t+1:t+l,p}^{\phi}(\cdot | y_{t-k+1:t,p})$ is defined as:

$$\phi_p(X, y_{t+1:t+l}) = \frac{1}{m(m-1)} \sum_{r \neq s} \left[ K^{\text{Sig}}(X_p^r, X_p^s) \right] - \frac{2}{m} \sum_{r=1}^{m} \left[ K^{\text{Sig}}(X_p^r, y_{t+1:t+l,p}) \right]$$

However, this formulation may be misleading as the sigkernel package calculates the gram matrix of $K^{\text{Sig}}(X, X)$ and $K^{\text{Sig}}(X, Y)$ instead of individual terms.

## 4.2 Minimum Prequential Signature Kernel Score Multistep forecaster for WeatherBench 2

Unlike in evaluation, some standard weather models, such as the ones in Table A.1, do not train on latitude weighted scores (Pathak et al., 2022)(Nguyen et al., 2023). As a result, we are not forced to take latitude slices and can consider global paths or patches. Patches also disrupt spatial structure, but include both latitude and longitudinal correlations and may capture local behaviour more effectively. Additionally, lower dimensionality is helpful to reduce numerical stability in this method, addressed further in Section 4.3. Both approaches were tested, and patches proved slightly more effective. We consider 24 patches of size 16x16, with three latitude and eight longitude initialisation positions, accounting for longitude wrapping. Each patch has a dimension size of 256, eight times smaller than the global 2048 dimensions. Patches have considerable overlap to ensure smooth prediction behaviour and cover the equivalent of three globe areas. Overlap naturally occurs in the non-polar regions, which implicitly latitude weights to a moderate degree.

We employ a U-NET architecture for the generative network, details of which is provided in the Appendix I. For training, we consider geopotential 500 hPa on the roughest resolution of 64 by 32 longitude-latitude regions. We use global data from 2010 to 2018, with a 6:2:1 train, validation, test split using the years 2010-2015, 2016-2017, and 2018, respectively. Optimal training would involve a significantly larger number of years to capture longer term weather patterns. However, these data choices were made to balance learning ability and computational time. We consider all experiments with a maximum of 100 epochs and validation early stopping in increments of 5 epochs past 20. We train across 4 path lengths (2, 5, 10, 15) and each across 9 learning rates, logarithmically spaced between 1e-2 and 1e-6. All use the RBF static kernel with $\sigma = 1$, dyadic order of 1, and initial context window size of 10. Maximum epoch time was never reached during training. The model chosen for evaluation is based on the lowest final validation loss reached. We consider a batch size of 16 patches due to memory constraints on the GPU. Experimental results with a one-step prequential scoring rule on WeatherBench data showed equivalent metrics of calibration error, NRMSE, and $R^2$ across training ensemble numbers of three to fifty (Pacchiardi et al., 2024). Therefore, we train with three ensemble members to minimise computation time with no compromise to performance. All metrics are evaluated on an evaluation ensemble size of 200.

Table 1 displays the latitude-weighted metric results of training on WeatherBench Data across four different path lengths. As expected, smaller path lengths result in stronger performance, particularly notable in NRMSE and $R^2$. At path length 15, the model's $R^2$ is roughly equal to zero, suggesting equal performance to mean climatology. Nonetheless, the results across path lengths are strong, with low NCRPS and low calibration error. For all path lengths, NCRPS is substantially less than 1, implying a forecast error smaller than the historical spread of climatology. Additionally, calibration error appears to plateau around 0.2 across path lengths 5 to 15.

In appendix I, we provide more figures illustrating the performance of the signature kernel trained models across path length. The discrepancies between ensemble members mostly exist in the temperate zone, where there is distinct boundary behaviour with polar regions, and errors are minimal for each initial forecast and build with increasing path length as expected.

## 4.3 Numerical Instability

The largest issue with training implementation is the instability of computation, with different scales and static kernels causing issues with Inf/Nan values or scores converging to 1. In multiple test instances, the signature kernel score appears to be calculating correctly for small prediction lags, but then displays

| Training Results for WeatherBench 2 (Geopotential 500 hPa) | | | | | | |
|---|---|---|---|---|---|---|
| Path L. | LR | Cal.Err. ↓ | NRMSE ↓ | $R^2$ ↑ | NCRPS ↓ | RQE |
| 2 | 0.001 | 0.0928 | 0.0947 | 0.6823 | 0.2908 | 1.4905 |
| 5 | 0.0003 | 0.1612 | 0.1441 | 0.2398 | 0.4568 | 1.5048 |
| 10 | 0.0003 | 0.2136 | 0.1645 | 0.1375 | 0.5462 | 1.4795 |
| 15 | 0.003 | 0.1886 | 0.1818 | -0.0160 | 0.5855 | 1.3975 |

Table 1: Evaluation metrics across path length for minimum prequential signature kernel score model for Geopotential 500 hPa from 2010 to 2018.

abnormal converging behaviour. Theoretical and experimental results were limited in application to this context. The generalised data augmentation method proposed by Morrill et al. (2021) demonstrated results of different approaches varying significantly by dimensionality of the path, with eight being the largest dimension tested. Moreover, results were only conducted for path length $l \gg d$, the dimension of the data. In this implementation, the dimensionality of the path is 2048 in evaluation and 256 in training.

This type of path is not covered by existing literature. Nonetheless, extending these results proved successful. Standard normalising using the mean and standard deviation of only the observation data is conducted to allow comparisons between models with equal scaling. A further scaling constant of $\frac{1}{\sqrt{I \cdot J}}$ is applied, which is one over the square root of the longitude and latitude dimensions. Despite no theoretical backing, and with pre-processing rescaling being primarily considered for the truncated signature instead (Morrill et al., 2021), this addition solved detectable instability issues for ensemble models and across increasing resolution, verified up to the standard 240 by 121 resolution. The RBF kernel is considerably more robust to numerical instability issues in both evaluation and training than other kernels tested. The linear identity kernel was initially desired for handling high dimensional time series without complex feature mapping, to simplify the interpretation of the kernel, but numerical instability issues remain in training. We believe this instability arises from two compounding factors: the choice of static kernel directly controlling the magnitude of the loss landscape that occurs when predicted paths deviate significantly from observations, and the Goursat PDE solver accumulating numerical error when path differences are large (Salvi et al., 2021). This issue is evident with the linear and polynomial kernel, see Appendix G, compared to the RBF, which was more robust via its exponential decay moderating the loss landscape. As a result, we note that for reasonable kernel choices, these instability issues are not present in the primary use of the signature kernel score as a diagnostic, where predicted paths remain closely tied to the distribution of observed weather.

## 5 Conclusion

We have proposed a novel scoring rule for use in weather model evaluation and probabilistic forecast training. We have demonstrated appropriate behaviour in evaluation, along with potential structural differences in the data uniquely captured by the signature kernel. Our training approach demonstrates strong performance for path lengths up to 15 timesteps, at which point NRMSE is equivalent to climatology. As demand for reliable, fast, and high-resolution predictions increases, the signature kernel combines both theoretical support and demonstrated performance to display potential in next-generation forecasting systems.

We highlight the following extensions to this work: first, greater investigation into handling numerical stability through scaling and static kernels will allow for more robust training results across learning rates and extending to higher dimensionality weather forecasts. Secondly, our current approach of probabilistic non-adversarial training is not common, and implementation and adaptation into current weather forecasting frameworks are necessary for an appropriate analysis of performance. A further discussion of extensions and limitations of this work is provided in Appendix K.

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

# A   Evaluation Metrics

Table A.1 displays a summary analysis of the scoring rules in use among eight of the leading ML data-driven weather forecasting models between 2022-2024, determined by inclusion in the Google WeatherBench 2 framework (Rasp et al., 2020). These notably include the models from the researchers at Google DeepMind (Graphcast, GCM), Nvidia (FourCastNet FCN), and Huawei (Pangu-Weather PGW).

| Paper / Scoring Rule | Prob. | RMSE | MAE | ACC | CRPS | SSR | Act. | Bias | RQE | Cal. | ROC | MPE | SEEPS |
|---|---|---|---|---|---|---|---|---|---|---|---|---|---|
| **FourCastNet** (Pathak et al., 2022) | ✓ | ✓ | ✓ | ✓ | | | | | ✓ | | | | |
| **GNN** (Keisler, 2022) | | ✓ | | | | | | | | | | | |
| **GraphCast** (Lam et al., 2023) | | ✓ | ✓ | | | | | | | | | | |
| **NeuralGCM** (Kochkov et al., 2024) | ✓ | ✓ | | | ✓ | ✓ | | ✓ | | | | | |
| **FuXi** (Chen et al., 2023) | ✓ | ✓ | ✓ | ✓ | ✓ | ✓ | | | | | | | |
| **Pangu-Weather** (Bi et al., 2023) | ✓ | ✓ | ✓ | ✓ | ✓ | ✓ | | | ✓ | | | | |
| **Pangu-Weather Op.** (Bouallègue et al., 2024b) | ✓ | ✓ | | | | | ✓ | ✓ | | | ✓ | ✓ | |
| **Stormer** (Nguyen et al., 2023) | ✓[a] | ✓ | | ✓ | | | | | | | | | |
| **AIFS** (Lang et al., 2024a) | [b] | ✓ | | ✓ | | | ✓ | | | | | ✓ | ✓ |

 Pangu-Weather Op. refers to the operational version of Pangu-Weather. Abbreviations: Prob. (Probabilistic), Act. (Activity), Cal. (Calibration).
[a] Diagnostics not evaluated on the ensemble version of Stormer.
[b] AIFS is in ongoing development for ensemble forecasting.

Table A.1: Summary of included scoring rules across nine significant ML-based weather prediction models (as identified by WeatherBench 2). Abbreviations: ACC (Anomaly Correlation Coefficient), CRPS (Continuous Ranked Probability Score), SSR (Skill Spread Ratio), RQE (Relative Quantile Error), ROC (Receiver Operating Characteristic), MPE (Mean Position Error), and SEEPS (Stable Equitable Error in Probability Space).

## A.1   RMSE

Latitude weighting is important for assessing the quality of weather forecasts on the globe, as predictions are made over a grid of latitude and longitude squares spaced by equal degree intervals. These regions are much larger around the equator than the poles, which means that the contribution of each square towards the score needs to be weighted by its size. A discrepancy in a small region should be weighted far less than in a region multiple times its size. As the size is determined by the choice of latitude intervals, size is normalised via latitude weighting. WeatherBench's RMSE, which follows the same setup as ECMWF and the World Meteorological Organisation (WMO), is provided as follows for a specific level and variable $v$, with $f$ as the model forecast, and $o$ as the ERA5 true observation:

$$\text{RMSE}_v = \sqrt{\frac{1}{TIJ} \sum_{t=1}^{T} \sum_{i=1}^{I} \sum_{j=1}^{J} L(j)(f_{t,i,j}^v - o_{t,i,j}^v)^2}$$

Where $T$, $I$, $J$ are the length of the time, longitude, and latitude dimension, and $L(j)$ is the latitude weighting factor at the jth latitude index:

$$L(j) = \frac{\sin \theta_j^u - \sin \theta_j^l}{\frac{1}{J} \sum_{j=1}^{J} (\sin \theta_j^u - \sin \theta_j^l)} \tag{2}$$

with $\theta_i^u$ and $\theta_i^l$ being the upper and lower latitude bounds for the grid cell with latitude index (center) $i$. Polar regions, which would require a different formula, are excluded from the 64 by 32 5.625° resolution ERA5 dataset, which is used for training.

For use in evaluation to standardise RMSE results across variables, we introduce the normalised root mean square error NRMSE (Pacchiardi et al., 2024):

$$\text{NRMSE}_v = \frac{\text{RMSE}_v}{\max_t\{o^v_{t,i,j}\} - \min_t\{o^v_{t,i,j}\}}$$

This normalisation approach relates error to the total spread of the variable via the range of the observations. This result is sensitive to outliers within the observation data, which could result in underestimations of error for a specific variable. Nonetheless, this metric is useful as an intuitive and interpretable evaluation of error in relation to spread. In an ensemble forecast, the ensemble RMSE can be found by averaging across each model within the ensemble, ignoring any ensemble structure. The RMSE is only strictly proper for deterministic point forecasts, as it can not assess the full predictive distribution. This formulation of RMSE involves time within the square root, which differentiates from other formulations used by some models.

## A.2 $R^2$

We include a secondary deterministic metric, the coefficient of determination $R^2$. For a given latitude and longitude coordinate $i, j$, $R^2$ measures how well predicted values explain the variance observed in the data, defined as:

$$R^2_v(i,j) = 1 - \frac{\sum_{t=1}^{T}(f^v_{t,i,j} - o^v_{t,i,j})^2}{\sum_{t=1}^{T}(o^v_{t,i,j} - \bar{o}^v_{i,j})^2}$$

where $\bar{o}^v$ represents the observation mean. $R^2$ in this formulation is bounded above by 1, when forecasts equal observations across all times, $f^v_{t,i,j} = o^v_{t,i,j}$, but is unbounded from below, due to the absence of an intercept in the neural network. $R^2 = 0$ suggests a deterministic model which does no better than predicting the mean for every observation, with $R^2 < 0$, therefore suggesting poor model fit. We include this metric in evaluation to complement RMSE in assessing accuracy of the forecast mean predictions.

## A.3 CRPS

We next consider standard probabilistic scoring rules, designed for ensemble forecasts. The continuous ranked probability score (CRPS), which is strictly proper, is particularly relevant for the probabilistic forecasting of rainfall, due to rainfall having a positive point mass at 0. While other scoring rules for continuous variables are restricted to predictive densities, CRPS instead uses predictive cumulative distributions (Gneiting & Raftery, 2007). For deterministic forecasts, the CRPS reduces to MAE. The standard formulation for CRPS for a variable $v$, with the cumulative distribution function of its probabilistic forecast F is as follows:

$$\text{CRPS}(F_{i,j,t}, o_{i,j,t})_v = \int_{-\infty}^{\infty}(F^v_{i,j,t}(y)) - 1\{y \geq o^v_{i,j,t}\})^2 dy$$

During model evaluation or training, we must calculate these scores empirically. Extending to an ensemble model with M predictions, the latitude weighted, $L(j)$, empirical CRPS can be calculated as follows, letting:

$$\|g\|_{t,v} := \frac{1}{IJ}\sum_{i=1}^{I}\sum_{j=1}^{J}L(j)|g^v_{t,i,j}| \tag{3}$$

we define:

$$\text{CRPS}_v := \frac{1}{TM}\sum_{t=1}^{T}\left[\sum_{m=1}^{M}\|f^{(m)} - o\|_{t,v} - \frac{1}{2(M-1)}\sum_{m=1}^{M}\sum_{n=1}^{M}\|f^{(m)} - f^{(n)}\|_{t,v}\right] \tag{4}$$

While CRPS has some features suited for climate data, there are still no spatial or temporal structures incorporated. This score, while still underused, is insufficient for the highly dependent data structures in weather variables. Nonetheless, the CRPS, and its multivariate extension of the energy score are considered the standard in probabilistic forecasting. During model evaluation we consider a further extension of the

CRPS, which is the normalised NCRPS:

$$\text{NCRPS}_v(i,j) = \frac{\text{CRPS}_v(i,j)}{\sqrt{\frac{1}{T}\sum_{i=1}^{T}(o_{t,i,j} - \bar{o}_{i,j})^2}}$$

We evaluate the NCRPS across specific positions, and normalise by the standard deviation of observations. This normalisation is used to assess the quality of the forecast across the scale of the data, along with adding greater interpretation to CRPS results. A NCRPS greater than 1 represents a forecast error larger than the typical variation, standard deviation, of the observations. NCRPS smaller than 1 means the forecast error is smaller than the typical variation, suggesting reasonable forecasting performance, especially as NCRPS decreases to zero. This relates the performance to the historical spread of climatology in the region, which will be roughly equal to 1.

### A.4  RQE

Relative quantile error, RQE, is relevant to assess how well a probabilistic forecast captures specific quantiles compared to the true observation distribution. As we are concerned with capturing the uncertainty of extreme events, we are interested in the tail ends of the distribution. Both FourCastNet and Pangu-Weather set $D = 50$ percentiles linearly distributed between 90% and 99.99% on the logarithmic scale, which is the approach adopted in this work as well. For each percentile, we determine the corresponding quantile calculated from the observational truth $Q_d$ and from the forecast $\hat{Q}_d$. Computed separately for each weather variable $v$, we solve the following:

$$\text{RQE}_v = \sum_{d=1}^{D} \frac{\hat{Q}_d - Q_d}{Q_d}$$

RQE measures the overall forecast tendency across all extreme quantiles, with $\text{RQE} < 0$ implying a trend of underestimation of extremes, while $\text{RQE} > 0$ implies an overestimation of extremes compared to the true distribution. However, this value is limited by not assessing whether extreme values are predicted correctly, as it is insensitive to correct alignment with observations. As a result, this value needs to be interpreted along with an assessment of forecast accuracy (Bi et al., 2023).

### A.5  Calibration

We consider the assessment of calibration for probabilistic forecasts to be a relevant measure for inclusion in evaluation. We use the definition from Radev et .al of calibration error as the difference between credible intervals in the forecast and observation distributions (Radev et al., 2020). For each variable, we consider 100 credibility levels $\alpha$ spaced evenly between 0.01 and 1. We compute the $\alpha/2$ and $1 - \alpha/2$ quantiles from our forecast samples $f^v$, $q_{a/2}^v$ and $q_{1-a/2}^v$ respectively. For each credible level we determine the proportion of true samples which lie within the corresponding credible interval, known as the empirical coverage or proportion of inliers:

$$\hat{\alpha}^v = \frac{1}{N}\sum_{j=1}^{N}\mathbb{I}(o_j^v \in \left[q_{a/2}^v, q_{1-a/2}^v\right])$$

The calibration error is then the median of all absolute deviations across credible levels:

$$\text{Calibration Error}_v = \text{Median}(|a_1 - \hat{\alpha}_1^v|, \ldots, |a_{100} - \hat{\alpha}_{100}^v|)$$

For a perfectly calibrated forecast, $\alpha = \hat{\alpha}$ for all credibility levels, resulting in a perfect calibration score of 0. If credible intervals are either too narrow or too wide, representing over and underconfidence, calibration error may increase with the median deviation. The median is used as an aggregating function to capture the systematic bias present in the deviation between coverage and credibility level. The mean deviation is not robust to outliers, and does not align with the frequentist goal of credible intervals to summarise the typical calibration. A calibration error of nearly 0.5 represents a fully miscalibrated forecast, which is extremely difficult to design intentionally, with every credible interval excluding the true parameter except for $\alpha = 1$.

# B  Geometric Signature Interpreations

Path integrals include self-referential integrations, with $i_1 = i_2$, referred to as diagonal terms, resulting in $d^2$ combinations of second order terms. Second order diagonal terms are a simple extension of first order terms with:

$$S(X)_{a,b}^{i,i} = \frac{(X_b^i - X_a^i)^2}{2} = \frac{(S(X)_{a,b}^i)^2}{2} \tag{5}$$

However, non-diagonal terms, referred to as cross terms, with $i_1 \neq i_2$ for second order terms are more complicated, relating to the Lévy area of the two-dimensional path. The Levy area is the signed area between a path and a chord created between its endpoints, represented in Figure B.1 (Chevyrev & Kormilitzin, 2025) with areas $A_-$ and $A_+$, denoted by:

$$A_{\text{Lévy}} = A_+ - A_- = \frac{1}{2}\left(S(X)_{a,b}^{i_1,i_2} - S(X)_{a,b}^{i_2,i_1}\right)$$

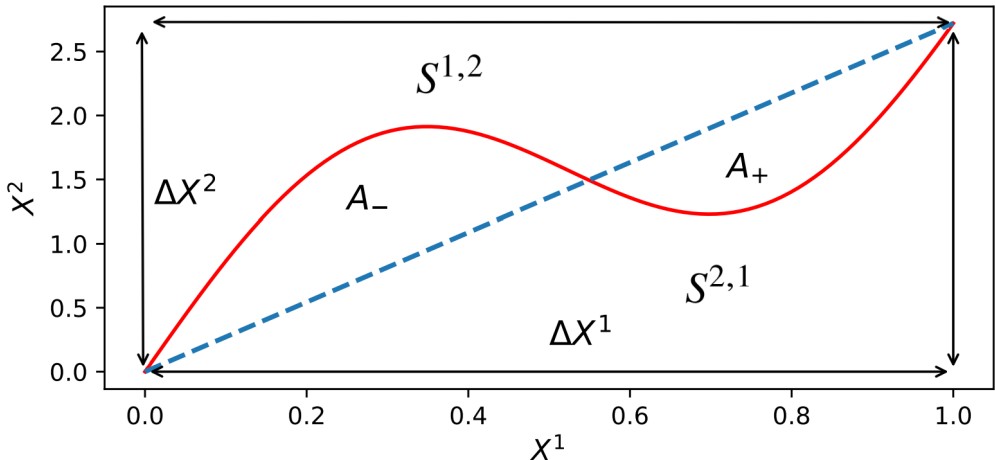

Figure B.1: Representation of the signed Lévy area of a two-dimensional path. $A_+$ and $A_-$ denote the positive and negative respective areas between the path (red) and chord (blue). $S^{1,2}$ and $S^{2,1}$ denote the second order terms equal to the area between the path and their perpendicular enclosure (black).

From the figure above we can observe a further connection between first and second terms, noting the sum of second order terms is equivalent to the area of the rectangle defined by side lengths of the first order:

$$S(X)^{i_1} \cdot S(X)^{i_2} = \Delta X^{i_1} \cdot \Delta X^{i_2} = S(X)^{i_1,i_2} + S(X)^{i_2,i_1} \tag{6}$$

We can extend the results found in Equations 5 and 6 using the shuffle product identity (Ree, 1958). This identity shows that we can relate the product of any two signature terms $S(X)_{a,b}^{i_1,\dots,i_k}$ and $S(X)_{a,b}^{j_1,\dots,j_m}$ as the sum of $S(X)_{a,b}^K$ terms, which depend only on indexes $(i_1, \dots, i_k)$ and $(j_1, \dots, j_m)$. This collection of new indexes, $K$, is obtained by permuting both index sets together while keeping their respective ordering, referred to as shuffling and denoted by $\sqcup\!\sqcup$. Given a permutation $\sigma$ of the set $\{i_1, \dots, i_k, j_1, \dots, j_m\}$, $\sigma$ is a $(k,m)$-shuffle if $\sigma^{-1}(i_1) < \dots < \sigma^{-1}(i_k)$ and $\sigma^{-1}(j_1) < \dots < \sigma^{-1}(j_m)$. We can therefore define the shuffle product identity:

$$S(X)_{a,b}^{i_1,\dots,i_k} \cdot S(X)_{a,b}^{j_1,\dots,j_m} = \sum_{K \in \{i_1,\dots,i_k\} \sqcup\!\sqcup \{j_1,\dots,j_m\}} S(X)_{a,b}^K$$

This summation occurs over $\binom{k+m}{k}$ shuffles, where $K$ is in general a multi-set which contains multiple identical shuffles for orders of the same index, e.g. (i) $\sqcup\!\sqcup$ (i) $= \{(i,i), (i,i)\}$. We can now generalise equation 5 for any k-order term with a single index with the following:

$$S(X)_{a,b}^{i,\dots,i} = \frac{(X_b^i - X_a^i)^k}{k!} = \frac{(S(X)_{a,b}^i)^k}{k!}$$

## C   Extension to Discrete Weather Data

Section 2.2 considered signatures defined for smooth paths. However, in our context, we would like to view our discrete time forecasts and observations as paths. Additionally, with handling real weather data, it is necessary to consider how irregularly sampled or partially observed data affects the result. This requires extending our definitions of the signature from paths to data streams consisting of discrete points. We consider a $d$-dimensional data stream, $X \in \mathbb{R}^d$ for $N$ points with an ordered set of times, $t_0 < t_1 < \cdots < t_N$, as follows:

$$X = \{(X_{t_i}^1, \ldots, X_{t_N}^d)\}_{i=0}^N$$

We then need to transform this datastream into piecewise paths, satisfying piecewise $C^1$ differentiability (Salvi et al., 2021). Multiple possible transformations exist to satisfy this condition. However, the linear interpolation approach is most compatible with rough path theory, conventional in signature implementation, and better captures curvature and orientation behaviour compared to other methods (Reizenstein & Graham, 2018). Therefore, for sequential pairs of points within the data stream, $X_{t_i}$ and $X_{t_{i+1}}$, we define the linear path between them as:

$$\hat{X}_t = X_{t_i} + \frac{t - t_i}{t_{i+1} - t_i}(X_{t_{i+1}} - X_{t_i})$$

Each linear interpolation is piecewise infinitely continuously differentiable $C^\infty$ and satisfies the requirements of a smooth path. The overall path is the result of concatenating each interpolated section, resulting in a continuous $C^0$ path for any $t \in [t_0, t_N]$ (Chevyrev & Kormilitzin, 2025). We now revisit the calculation of the signature by considering signatures of each interpolated linear path on their respective interval $[t_i, t_{i+1}]$. Letting $x$ be the straight line defined on $[t_i, t_{i+1}]$, we define the signature:

$$S(X)_{t_i:t_{i+1}}^{j_1,\ldots,j_k} = \frac{1}{k!} \prod_{l=1}^k (x_{j_l}(t_{i+1}) - x_{j_l}(t_i))$$

This result can be understood through the similarity to the diagonal terms of a smooth path signature, as the path between the same index terms is a linear path. Defining $\gamma$ as $x(t_{i+1}) - x(t_i)$, and grouping the signature by order levels, we can solve for each level of the full signature:

$$\left(1, \gamma, \frac{\gamma \otimes \gamma}{2!}, \frac{\gamma \otimes \gamma \otimes \gamma}{3!}, \ldots\right) \tag{7}$$

For a specific dimension index combination, we consider $\otimes$ the tensor product. If we consider the appropriate vector of numbers for each level, $\otimes$ can be taken as the Kronecker product across all variable combinations (Reizenstein & Graham, 2018). To consider the full signature of the overall path, we employ Chen's identity (Chen, 1954). Considering three time points in the datastream satisfying $t_a < t_b < t_c$ we can write:

$$S(X)_{t_a:t_c}^{j_1,\ldots,j_k} = \sum_{l=0}^k S(X)_{t_a:t_b}^{j_1,j_2,\ldots,j_{l-1}} \cdot S(X)_{t_b:t_c}^{j_l,j_{l+1},\ldots,j_k}$$

We can once again group the signature by order levels and consider the first four depths, this time considering the overall path:

$$S(X)_{t_a,t_c}^1 = S(X)_{t_a,t_b}^1 + S(X)_{t_b,t_c}^1$$

$$S(X)_{t_a,t_c}^2 = S(X)_{t_a,t_b}^2 + S(X)_{t_a,t_b}^1 \otimes S(X)_{t_b,t_c}^1$$

$$S(X)_{t_a,t_c}^3 = S(X)_{t_a,t_b}^3 + S(X)_{t_a,t_b}^2 \otimes S(X)_{t_b,t_c}^1 + S(X)_{t_a,t_b}^1 \otimes S(X)_{t_b,t_c}^2 + S(X)_{t_b,t_c}^3$$

$$S(X)_{t_a,t_c}^4 = S(X)_{t_a,t_b}^4 + S(X)_{t_a,t_b}^3 \otimes \cdots$$

Now the signature is defined for the region $[t_a, t_c]$, and can be extended to further points $t_d$ with step by step concatenation following the same method. We now have a method to consider weather data and our associated forecasts as paths which can be appropriately transformed into a signature, capturing the dependencies of our chronological sequence across the latitude-longitude grid.

## D   Further Signature Augmentations

There are a variety of other path augmentations which are often considered. Lead-lagging the data is one of the most common augmentations, which greatly increases the dimension size by considering both a lagged and leaded path of the original data. This augmentation can be represented as the following:

$$\phi(\mathbf{x}) = ((x_1, x_1), (x_2, x_1), (x_2, x_2), (x_3, x_2), \ldots, (x_n, x_n)) \in \mathcal{S}(\mathbb{R}^{2d})$$

As the signature extracts information about enclosed areas with the path, this formation directly aligns with this ability by causing loops to appear from the data. This representation benefits the capturing of certain statistics, particularly the quadratic variation, which can be represented as signature terms of the lead-lagged path (Chevyrev & Kormilitzin, 2025):

$$A_{\text{Lévy}}^{(X^{\text{Lead}}, X^{\text{Lag}})} = \frac{1}{2} \sum_{i=1}^{N} (X_{t_i} - X_{t_{i-1}})^2$$

As a result, the choice to lead-lag may depend on both the data and forecasting goals. However, existing evidence suggests poorer performance on a variety of datasets compared to not lead-lagging, particularly for higher dimensional data with $d > 8$ (Morrill et al., 2021). Weather data is incredibly high dimensional, as each latitude-longitude region needs to be considered as a dimension, and this augmentation was not included in implementation.

An additional choice of windowing operation can be taken. The global window is considered naturally, but changing window size via sliding, expanding or hierarchical dyadic windows changes how data is read to capture information at different scales. There is evidence that the hierarchical dyadic windowing outperforms the global window (Morrill et al., 2021). However, during implementation we observed that large path lengths posed challenges for any strategy of weather forecasting. Over the smaller path lengths that were considered, the global window was deemed suitable. Further investigation into this hyperparameter choice would be a valid extension.

Rescaling in this context is generally considered as a post processing method to be applied to the terms in the signature. As can be observed in Equation 7, the depth-k terms of the signature are of size $\mathcal{O}(\frac{1}{k!})$. Rescaling these terms to $\mathcal{O}(1)$ by multiplying each depth by $k!$ may help ensure these features are addressed. Pre-signature scaling involves multiplying an input by a scaling factor $\alpha \in R$. The depth k terms will be of size $\mathcal{O}(\frac{a^k}{k!})$. However, no theoretical justification exists for optimal $\alpha$ values, and issues arise as different depth-k terms cannot be adjusted separately. Both methods suffer greatly from numerical stability issues. However, for reasons explained in the following section, only pre-scaling could be considered.

The final common augmentation choice is between using the standard signature or logsignature transformation. A logsignature lowers the feature dimension by removing some of the redundancies in the existing signature, such as the relationship in Equation 6 connecting the first and second order terms together. However, as a result, it loses linear approximation properties, called universal nonlinearity (Bonnier et al., 2019). Given the function $f$ which maps data $\mathbf{x}$ to labels $y$, the universal nonlinearity property, with some assumptions, declares that for any $\epsilon > 0$, there exists a linear function $L$ such that:

$$||f(\mathbf{x}) - L(S(\mathbf{x}))|| \leq \epsilon$$

The standard signature provides a natural basis for functions of the time series. In practice, there is no consensus on which version is most suited for a machine learning task. Additionally, as shown with the kernel trick, we do not need to be concerned with lowering the feature dimension of the signature, and so we use the standard signature.

## E   Strict Propriety

*Proof of Theorem 1.* The score in Equation 1 takes the standard form of a kernel score, which is proven to be strictly proper on a compact set of paths $\mathcal{X}$ if and only if $K^{Sig}$ is a characteristic kernel on that space

(Steinwart & Ziegel, 2021, Theorem 1.1). We therefore need to verify **(i)** the compactness of the path space, and that **(ii)** the signature kernel is characteristic.

**(i) Compactness of the path space.** In the discrete forecasting setting, paths are piecewise linear with fixed knots at times $0 = t_0 < \cdots < t_N = T$ and values in a compact spatial domain $K \subset \mathbb{R}^d$ (closed and bounded via physical properties of weather variables), so the path space is:

$$\mathcal{X} = \{x \in C([0,T], \mathbb{R}^d) : x \text{ is linear on each } [t_k, t_{k+1}], \ x(t_k) \in K \ \forall k\}.$$

We verify the hypotheses of the Arzelà–Ascoli theorem (Rudin, 1976, Thm. 7.25).

1. *Uniform boundedness* is immediate since every path takes values in $K$.

2. *Equicontinuity* follows because each path is Lipschitz with the shared constant $L = \text{diam}(K)/\Delta t_{\min}$, where $\Delta t_{\min} = \min_k (t_{k+1} - t_k)$: on every linear segment $|x(t) - x(s)| \leq L|t - s|$, and the bound extends globally over segments by the triangle inequality.

3. *Closedness* holds because uniform limits of piecewise-linear paths with knots at $t_0, \ldots, t_N$ and values in the closed set $K$ retain both properties.

Thus $\mathcal{X}$ is compact in the uniform topology.

**(ii) Characteristicness of the signature kernel.** The signature kernel, induced by a characteristic static kernel (such as the radial basis function kernel), is a characteristic kernel on compact sets of bounded-variation paths provided the paths are distinct (Issa et al., 2023, Props. 3.1, 3.3). Bounded variation holds for every $x \in \mathcal{X}$, since each path is Lipschitz. Path uniqueness is guaranteed by the applied basepoint and time signature augmentations (Section 2.3), which removes spatial and temporal invariances.

**Conclusion.** Conditions (i) and (ii) together satisfy the hypotheses of Steinwart & Ziegel (2021, Theorem 1.1), so the signature kernel score (1) is strictly proper, exhibiting the desired fundamental property of a scoring rule, ensuring truthful predictions. $\square$

## F   Evaluation Implementation

There are two different implementation approaches taken for either evaluation or model training, both relating to latitude weighting and stability of the signature kernel. Training implementation and the discussion of generative model architecture and the prequential are found in the main body in Section 4.

Here we address evaluation, where latitude weighting is critical for appropriate model assessment. Including spatial correlations within the signature kernel score while latitudinally weighting requires additional considerations. Latitude weighting is necessary to prevent the poles from having a disproportionate contribution to the model compared to the equator. However, due to the non-linear transformation of the signature and the implicit mapping via the kernel, different latitude regions can't be appropriately pre-weighted before applying the signature kernel. Instead, from each initialisation time, multiple paths are considered, each corresponding to a latitude slice with dimensionality equal to the number of longitude positions at every time step. From each path, the relevant values to calculate the score or distance are determined between the forecasted data X and observations Y: $K^{Sig}(X, Y), K^{Sig}(X, X), K^{Sig}(Y, Y)$. A weighted average is then able to be conducted on these values associated with each latitude slice, before calculating the score or distance. This requires a revisit to the signature kernel definitions. In the simpler deterministic case, we consider a forecast $F_{i,j,t,k}$ with longitude, latitude $i, j$, with respective number of each $I, J$, the set of initialisation times $\{t\}$, and future predictions $1 : k$ for each initialisation time, along with observations $o_{i,j,t}$, with likewise $i, j$ and values at $\{t\} + 1 : k$. We consider the latitude weighted signature distance and score for a path length of $k \geq 2$ over the set of initialisation times $\{t\}$, as follows:

$$d^2(K_{k,t}^{Sig}) = \sum_{j=1}^{J} w_j \sum_t d^2(F_{1:I,j,t,0:k}, o_{1:I,j,t:t+k})$$

$$\phi(K_{k,t}^{Sig}) = \sum_{j=1}^{J} w_j \sum_{t} \phi(F_{1:I,j,t,0:k}, o_{1:I,j,t:t+k})$$

Variables measured at different atmospheric pressures, commonly 500 hPa and 850 hPa, which correspond to differing three-dimensional areas, are treated independently. The summation over longitude slices removes the spatial correlations between different latitudes from being captured by the signature kernel during evaluation. However, without latitude separation, regions are not compared appropriately. Hemisphere and tropical separations, which are required for scorecard analysis, are created by assigning the appropriate latitude slices.

## G  Discriminative Power

Discriminative power is the ability of a score to distinguish between different forecast and observation paths. High discriminative ability means the score gives significantly different values to forecasts of different quality. Highly discriminative scores prevent ambiguous results when comparing models and also greatly affect the sensitivity of the loss landscape during training (Pinson & Tastu, 2013).

To investigate discriminative power, we implement a stochastic vegetation and fire spread model. Given a two-dimensional grid, we define the vegetation and heat at time $t$ in the coordinate $(i, j)$ as $V_{i,j}^t$ and $H_{i,j}^t$ respectively. Every time step has the following three processes: vegetation growth, heat spread, and vegetation consumption, using the rates for vegetation growth,$a$, ignition sensitivity, $b$, combustion, $c$, fire day, $d$, noise scale, $s$, and time step $\Delta t$. We introduce stochasticity into the model with $\epsilon, \zeta \sim \mathcal{N}(0, s^2)$.

$$V_{i,j}^{t'} = V_{i,j}^t + a(1 - V_{i,j}^t)\Delta t + \epsilon_{i,j}^t$$

$$H_{i,j}^{t+1} = H_{i,j}^t + (b \cdot \bar{H}_{i,j}^t \cdot V_{i,j}^{t'} - dH_{i,j}^t)\Delta t + \zeta_{i,j}^t$$

$$V_{i,j}^{t+1} = V_{i,j}^{t'} - cH_{i,j}^{t+1}\Delta t$$

$\bar{H}_{i,j}^t$ is the average value of distance-weighted adjacent and diagonal squares. As a result, we have a simple model, which we can easily simulate over a large grid, with temporally and spatially complex relationships. Vegetation can be thought of as primarily temporally related, while heat is both spatially and temporally correlated. This model was chosen for low time complexity generation to enable scores over large numbers of ensembles, along with smooth path extensions given any initial input data.

To investigate discriminative power, we consider the following experiments on the particular variables of vegetation growth and fire spread intensity. We consider a dense spectrum of changes to each parameter, and for each shifted value, we create the paths of 50 ensemble members, along with 100 simulations of true path trajectories with an unaltered true parameter. As a result, we can estimate the expected score for each ensemble set associated with a differently valued parameter. For a given parameter, ensemble members result in moderately varied path trajectories via the stochasticity in each time step. We assess the relative discriminative power of the signature kernel as compared to the energy score, average MSE, and the variogram score.

We investigate these scores on changes to vegetation growth in Figure G.2. The x-axis represents the change to the shifted parameter of the particular ensemble set. Therefore, the minimum score is expected at zero when both the ensembles and observation paths have aligned parameters. We find that each score captures this variation similarly, with nearly identical curvature up to differences in scale.

We consider these scores on changes to fire spread intensity in Figure G.3. On this parameter, which is both spatially and temporally correlated, we see the signature kernel score displaying a much more drastic spike in score compared to the alternatives. We additionally see greater increases in score with positive changes to fire spread intensity compared to flattening with negative changes. This is intuitive, as the fire dies out for a range of lower fire spread values, causing similar paths, while the opposite is true for fire propagation. The asymmetric profile of the signature kernel score reflects the physical structure of the model. However, despite high sensitivity in the positive region, avoiding plateaus is particularly relevant for learning, i.e. training a

model, as our predicted forecasts need to be directed to the optimum. Despite sharp discriminative ability in comparing similar forecasts near the optimum, i.e. the case of diagnostic evaluation between two well trained models, the plateau regions could result in slower, less optimal learning (Goodfellow et al., 2016). On the other hand, while the MSE and the variogram score display smoother curves, they are relatively flat in the region of the optimum, which makes them less sensitive as a diagnostic tool (Gneiting & Raftery, 2007).

In Figure G.4, we compare the discriminative behaviour of the signature kernel score across three static kernels, the RBF, the linear, and the polynomial kernel $d = 2$. Both the linear kernel and the polynomial kernel explode in value, with the polynomial kernel in particular displaying highly unstable behaviour. These suffer from numerical stability with unbounded growth and require significant attention to scaling. While asymmetrical, the RBF kernel score does not explode, with kernel values being bounded by $(0, 1]$ by construction. The exponential decay of the RBF kernel makes the behaviour observed qualitatively different from the overflow of the linear and polynomial kernels, with this kernel being the most robust to scale, and most suitable for training.

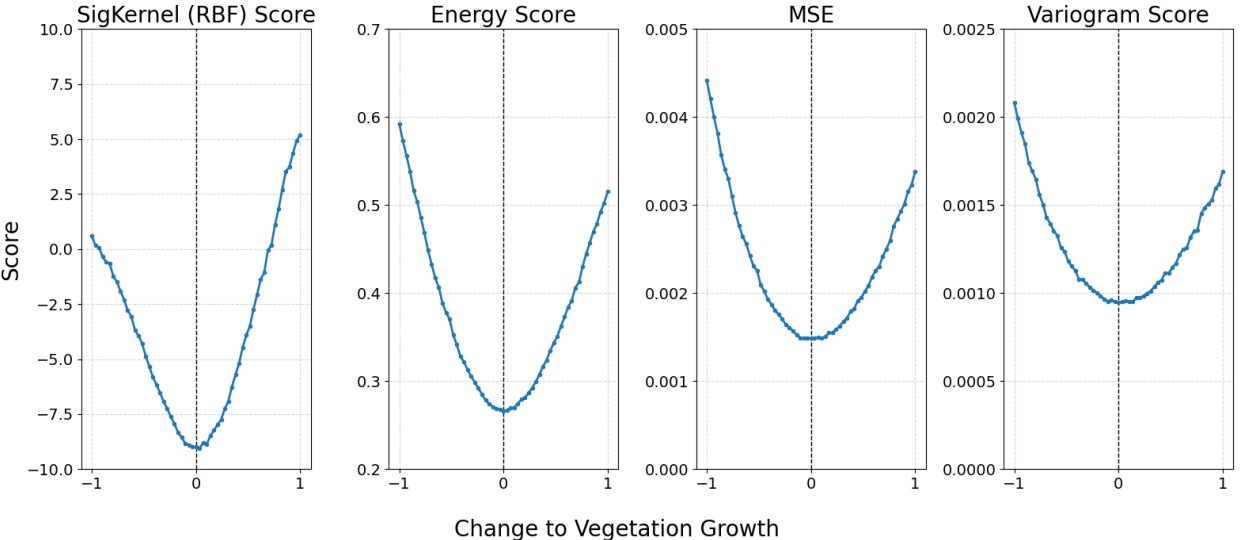

Figure G.2: Comparison of four scores across changes to the vegetation growth parameter. The vertical dotted line at zero represents the true parameter value, with negative and positive deviations to the left and right, respectively.

## H  WeatherBench data

WeatherBench 2 is an open-source evaluation frame-work which contains the current state-of-the-art models along with training, ground truth, and baseline data (Rasp et al., 2020). All models compared in this work originate from the WeatherBench 2 system. There are two distinct weather datasets considered by WeatherBench models, either Analysis or ERA5 reanalysis data. Both are created via the ECMWF's 4D-VAR data assimilation as best guesses of the true state of the atmosphere. However, Analysis is created in real time as the operational best predictor of the atmosphere. ERA5 reanalysis is a dataset created in retrospect, combining observations and ECMWF's high resolution (HRES) model predictions over twelve hour windows. In general, ERA5 data can be assumed to be closer to the truth than Analysis data, but operational models benefit from training with Analysis data. In either case, each model should be evaluated against the respective data set it was trained on, with significant differences only present for early lead times (WeatherBench, 2023). For this work, we have only considered ERA5 reanalysis data.

ERA5 data exists for each hour since 1940, however all current models only train on data since 1979. This year marks a significant jump in accurate weather observation, particularly for the Southern Hemisphere,

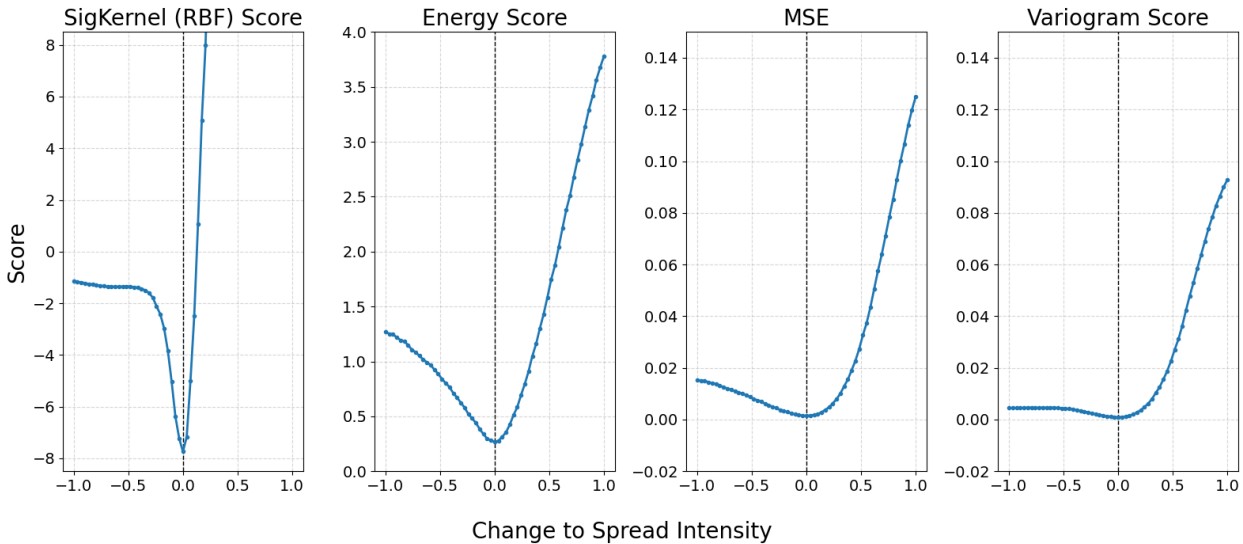

Figure G.3: Comparison of four scores across changes to the fire spread parameter. The vertical dotted line at zero represents the true parameter value, with negative and positive deviations to the left and right, respectively. The signature kernel score displays higher discriminative power than the other three.

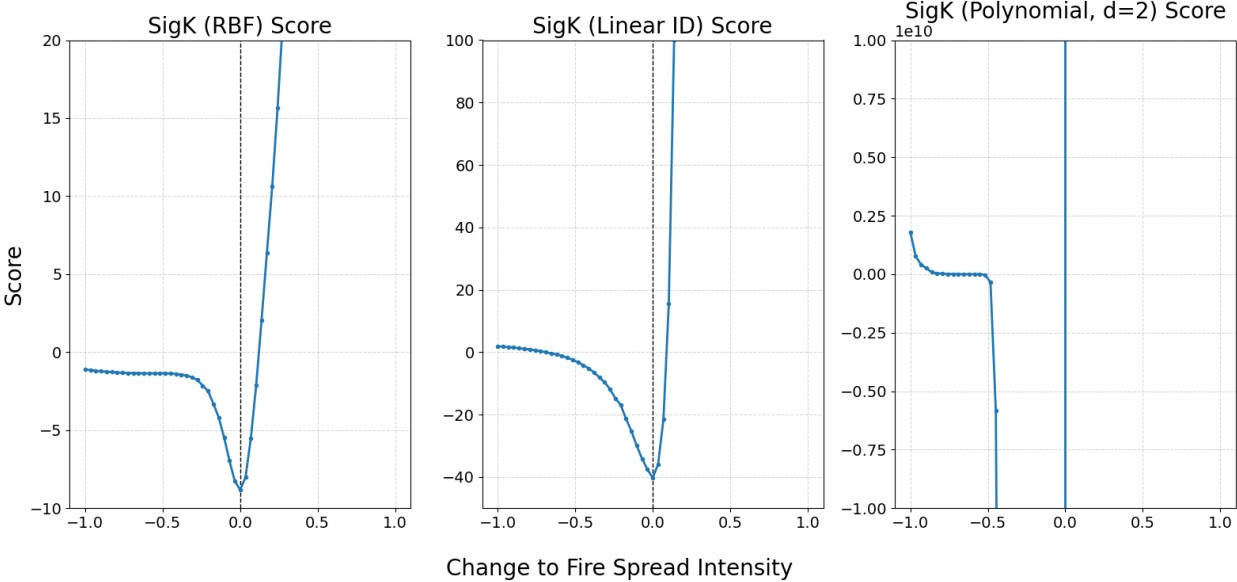

Figure G.4: Comparison of three static kernels across changes to the fire spread parameter. The vertical dotted line at zero represents the true parameter value, with negative and positive deviations to the left and right, respectively.

relative to previous years, as global satellite weather monitoring was introduced (Uppala, 2007). Additionally, most models consider a down-sampled version, with data every six hours at 00:00, 06:00, 12:00, and 18:00 UTC. Forecasts are initialised either on this schedule, or on a twelve hour cycle 00:00, 12:00. Predictions are then given every six or twelve hours. Forecasts are split between providing a zero-time initialisation or not, i.e. giving a prediction for its initialisation time. When comparing models with different zero-time forecasting approaches, this forecast was removed.

ERA5 reanalysis data between 1979 and roughly three months behind the present exists in completion without missing data at a 0.25° latitude-longitude coordinate resolution. It can be downloaded in its entirety from the Copernicus Climate Data store (Hersbach et al., 2023). Due to the extensive work in creating this dataset, there is no need to address any data integrity issues. Predictions from WeatherBench models can be downloaded from the WeatherBench platform. Certain models do have results missing for certain variables or resolutions, along with missing predictions. Variables or resolutions with missing data were not considered. The signature kernel method can handle missing data via the time augmentation, and so no further action was taken.

# I   Generative model architecture of multi-step probabilistic forecaster

We employ a U-NET architecture for the generative network. U-NET consists of encoders, which are a series of convolution layers followed by max pooling. Each level increases the number of feature maps, allowing the model to learn complex features. A decoder follows, which upsamples the features along with concatenating the results of previous encoders using skip connections. This allows the decoder to reuse features from earlier layers, preserving both low-frequency and high-frequency information. Additionally, we consider a bottleneck layer between the encoder and decoder where the latent variables are introduced, to make this process generative. We consider a U-NET architecture with depths of 32, 64, 128, and 256, respectively, illustrated in I.5. We implement the same window size of 10 previous time steps, and apply the same sliding window prequential generation. Therefore, the U-NET takes a time input of size 10, and gives an output for size 1, across the 64 by 32 spatial coordinates.

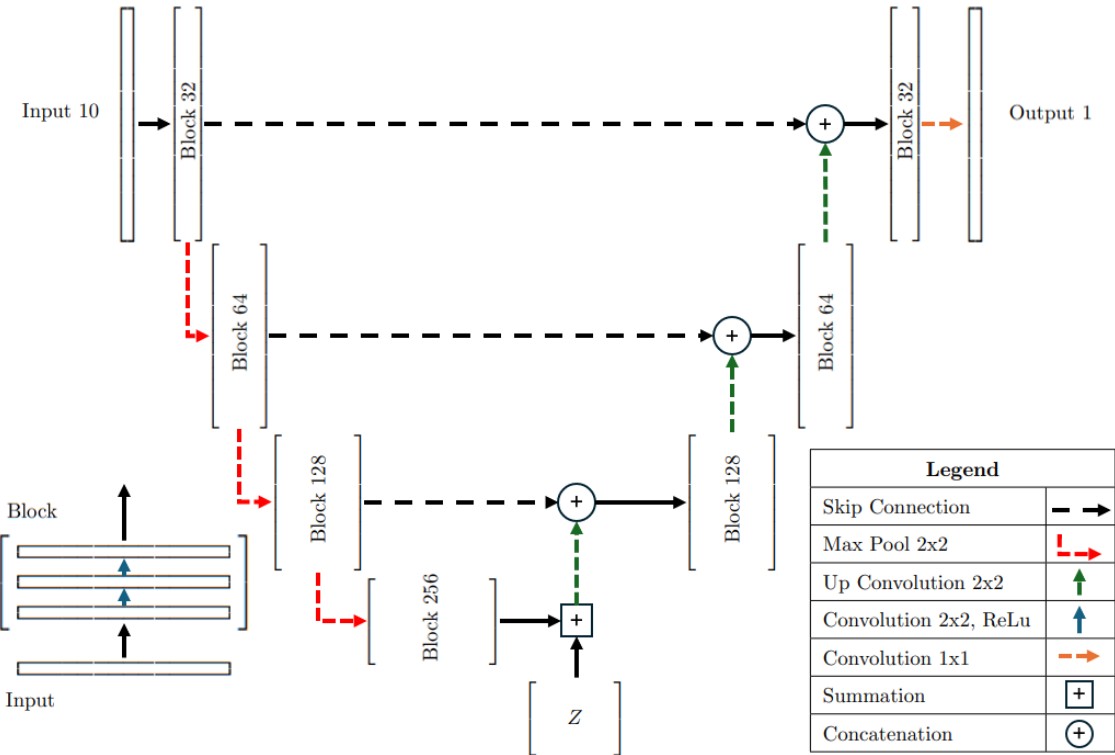

Figure I.5: Diagram of the implemented U-NET model architecture. The input size is equivalent to the window size of 10. The output size is 1. The U-NET has depths of 32, 64, 128, and 256. The latent variables, introduced between the encoder and decoder, are denoted by Z.

## J   Additional results for training multi-step probabilistic forecaster

The following figures illustrate the performance of the signature kernel trained models across path length. Figures J.6 and J.7 showcase ensemble spread for one realisation, with random variations between members to capture the forecast distribution. Most discrepancies exist in the temperate zone where there is distinct boundary behaviour with polar regions. Remaining figures showcase one ensemble member across respective path lengths of 5, 10, and 15. Errors are minimal for each initial forecast and build with increasing path length as expected.

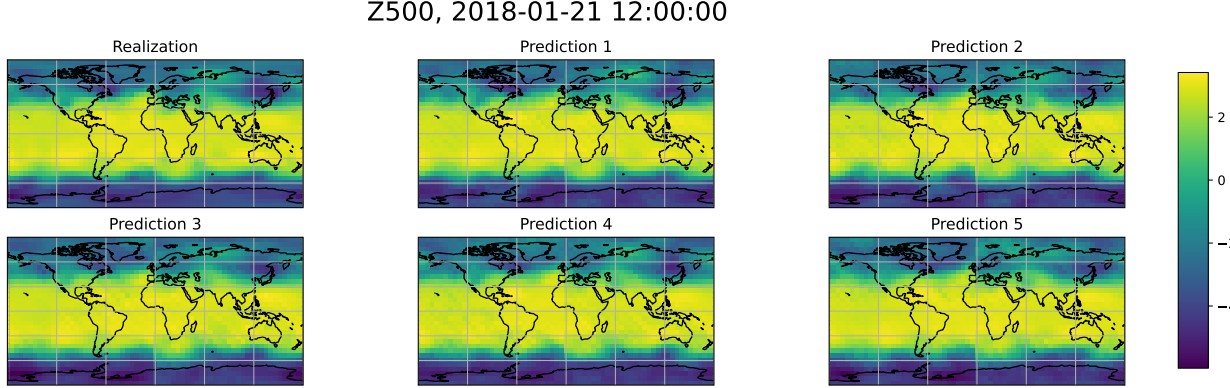

Figure J.6: Ensemble predictions for one initialisation generated from the path length 2 model.

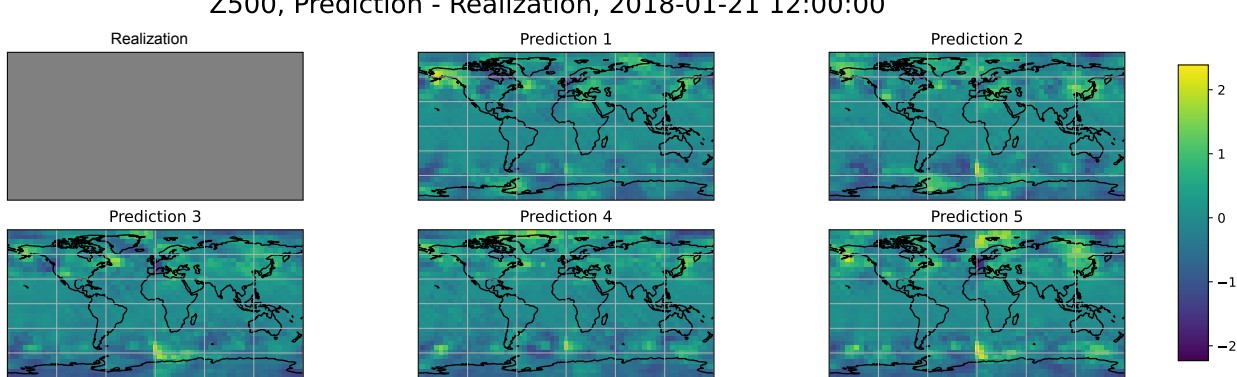

Figure J.7: Ensemble differences for one initialisation generated from the path length 2 model.

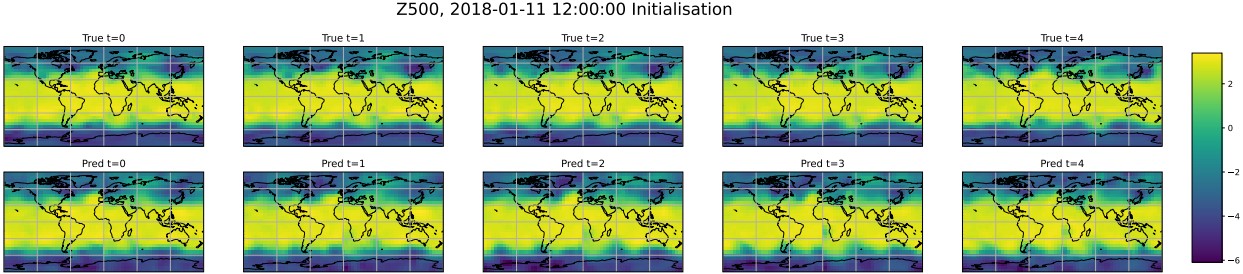

Figure J.8: Path comparison for one ensemble member generated from the path length 5 model.

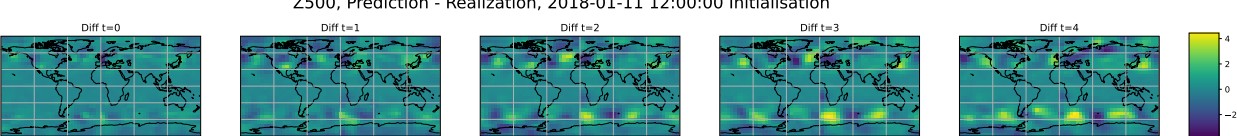

Figure J.9: Path differences for one ensemble member generated from the path length 5 model.

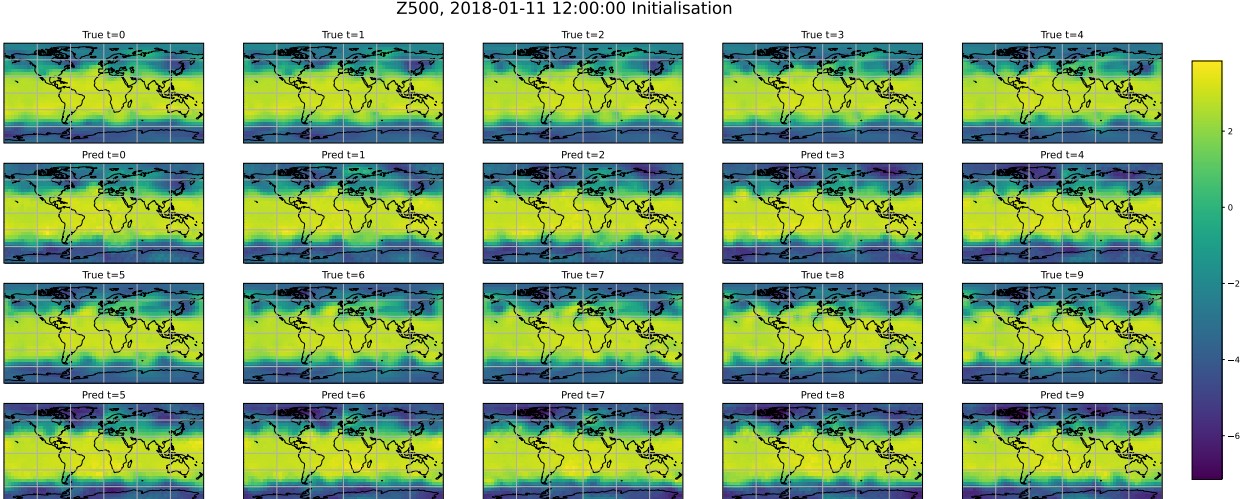

Figure J.10: Path comparison for one ensemble member generated from the path length 10 model.

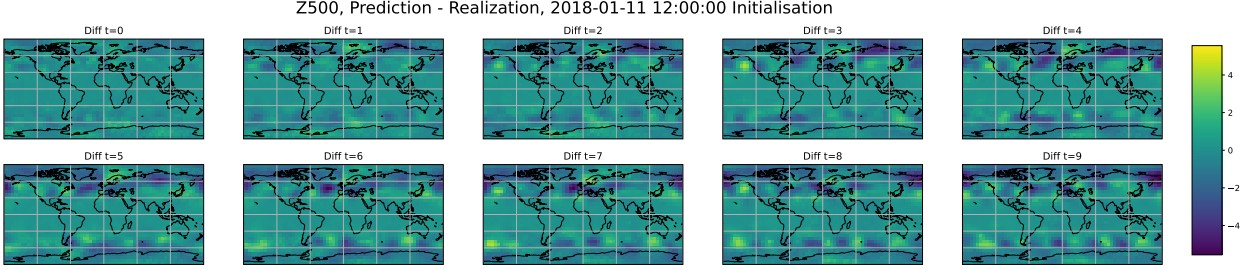

Figure J.11: Path differences for one ensemble member generated from the path length 10 model.

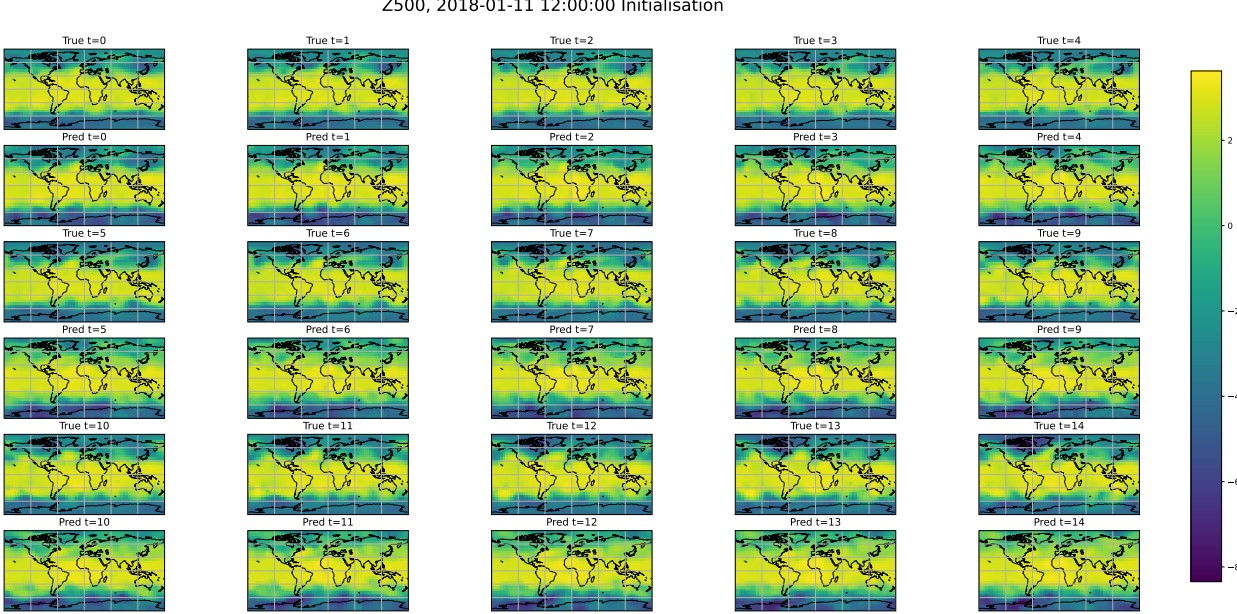

Figure J.12: Path comparison for one ensemble member generated from the path length 15 model.

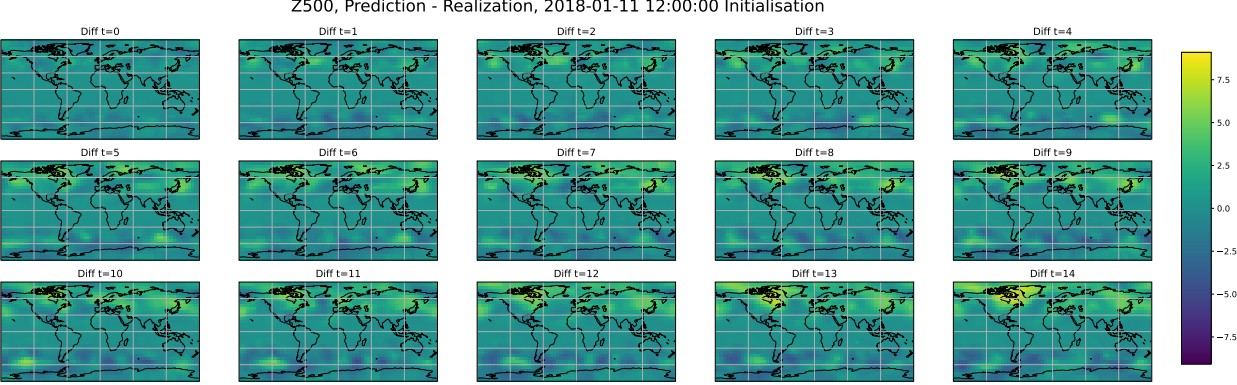

Figure J.13: Path differences for one ensemble member generated from the path length 15 model.

## K    Limitations

Several limitations of the current work have been collected below and are directions for future research.

As discussed in Section 4.3, numerical instability remains the one of the largest practical constraints on this method. Existing theoretical guidance on scaling and data augmentations has only been validated up to path dimensions of $d = 8$, while this work considers dimensions of up to 2048 (Morrill et al., 2021). Instability has been resolved in the tested settings using an empirically founded scaling fix, and the RBF static kernel with $\sigma = 1$, which may not be the most appropriate for the weather setting. Additionally, the choice of augmentation hyperparameters, such as window type and the choice not to lead lag, is informed by existing empirical evidence rather than optimised for weather data. Investigating the role of these choices would be a significant and valuable extension to this work.

The Goursat PDE solver used to compute the signature kernel requires dense signatures, meaning paths where signature terms are generally non-zero (Salvi et al., 2021). Specific weather variables, such as rainfall, are zero inflated and produce sparse signatures. This restricts the current implementation to variables with dense path behaviour, such as geopotential and temperature, but the extension to precipitation remains an open problem.

Spatial correlations are still limited by the problem of incorporating latitude weighting. The globe must be decomposed into smaller regions, either latitude slices or patches, in order to appropriately weight the signature kernel score. This necessarily discards cross-latitude or global spatial correlations. As a result, the signature kernel score can not fully exploit the global spatial structure that the method is theoretically capable of encoding.

The training scope is a further limitation. Training experiments are conducted exclusively on geopotential at 500 hPa at the coarsest WeatherBench resolution of 64 by 32, with a lightweight model. Nonetheless, with our primary aim to propose the signature kernel score as a diagnostic, we have demonstrated the reasonable success of our approach. An extension to benchmark our approach with state-of-the-art models, higher resolution data, and multi-variable forecasting would be required to draw conclusions on the method's performance for training in real-world scenarios.

Finally, we make assumptions of stationarity and mixing conditions for the underlying process to guarantee convergence for the prequential scoring rule minimiser (Pacchiardi et al., 2024). Weather data exhibits non-stationary behaviour with seasonal cycles and longer term climate trends, which means these assumptions may only be roughly satisfied. The practical impact of this violation has not been formally evaluated in this setting.

