# OpenReview forum: "Signature Kernel Scoring Rule: A Spatio-Temporal Diagnostic for Probabilistic Weather Forecasting"
_TMLR — Accepted by TMLR_

### Review · Reviewer_UqYp · 2025-12-10

**Summary Of Contributions:**

The authors introduce the "Signature Kernel scoring rule," a new metric for evaluating and training probabilistic weather forecasting models. Instead of treating weather predictions as isolated points in time (like MSE does), this method uses rough path theory to treat them as continuous paths, capturing the complex spatial and temporal dependencies typical in weather systems.

They validate that this rule is "strictly proper" (meaning it encourages truthful predictions) by using path augmentations. They put it to the test in two ways:

Evaluation: They created scorecards for existing WeatherBench 2 models (like NeuralGCM and IFS), showing that the signature kernel can pick up on performance nuances that standard metrics might miss.

Training: They trained a lightweight generative model (U-Net) using this scoring rule as a loss function, showing it can successfully learn to beat climatology for forecasts up to 15 steps ahead.

**Audience:**

Yes

**Audience Explanation:**

Definitely. The intersection of Machine Learning and Weather Prediction (MLWP) is a massive topic right now.

**Broader Impact Concerns:**

I don't see any major ethical concerns here. Improving weather forecasting generally has a positive societal impact. The authors briefly mention the standard broader impacts of weather forecasting, which seems sufficient.

**Claims And Evidence:**

Yes

**Claims Explanation:**

The paper does a solid job backing up its two main claims.

Metric properties: They provide the theoretical justification for the signature kernel being strictly proper and effective at capturing dependencies.

Practical utility: They show clear "weather scorecards" comparing the new metric against standard ones (ACC, MSE, CRPS) for real models. The results show the metric behaves reasonably and offers sharper discrimination in some cases (like FUXI in the Southern Hemisphere).

Training feasibility: They successfully trained a model using the prequential scoring rule framework. While the model is simple, the evidence supports the claim that this scoring rule can be used as a training objective. They are transparent about the model being lightweight and the comparison being against climatology.

**Requested Changes:**

- The numerical instability issues mentioned in Appendix C  are quite critical for anyone trying to implement this. I'd love to see a bit more of that discussion moved to the main text or at least highlighted more prominently in the implementation section. It's a practical hurdle that users need to know about upfront.
- Since the method involves solving Goursat PDEs, it would be helpful to have a clearer comparison of the computational cost (training time/inference time) of this loss function versus standard CRPS or MSE. Is it significantly slower to train?

---

> ### Author Response · Authors · 2026-03-02
>
> Thank you very much for your patience during the revision process. We have submitted a revised version of the document to address your recommendations. Along with the full list of changes, we have enhanced and moved the section on numerical instability into the main body (Section 4.3), along with an adjustment in Section 2.4 to discuss the time complexity implications during training.
>
> We hope you find that these additions strengthen the quality of the paper, and we welcome any further feedback.

---

### Review · Reviewer_z8ie · 2025-12-24

**Summary Of Contributions:**

The authors propose the signature kernel scoring rule (SIGK), a scoring rule designed for training and evaluating weather forecasting models. Intuitively, SIGK is an MMD with a custom kernel. The kernel is defined on signatures of augmented paths, which represent the time-discretized forecasts/observations of a single weather variable over a 2d spatial grid (latitudes x longitudes). Because the signature uniquely characterizes the (augmented) path, the authors claim that the associated scoring rule can better capture temporal and spatial dependencies relative to existing metrics like RMSE, which simply averages over the time and spatial points. The empirical validations are twofold: (1) through weather scorecards on the Weather Bench 2 models, they qualitatively show that SIGK as an evaluation metric is sensitive to spatiotemporal structures in ways that existing metrics aren't, and (2) uses SIGK as a prequential loss to train a generative forecaster on the ERA5 renalysis weather data.

**Additional Comments:**

Typos
- p4: For any dimension $i$ of our $d$-dimensional path $X : [a,b] \to \mathbb{R}$  (should be $\mathbb{R}^d$)
- p11: "unanticpated"

**Audience:**

Yes

**Audience Explanation:**

As written, this paper will be of interest to those training or evaluating weather forecasting models, but a few changes (requested below) will improve clarity and adoption.

**Claims And Evidence:**

No

**Claims Explanation:**

- Theoretical claims: The authors claim that SIGK is "validated as strictly proper through the use of path augmentations" but the paper would benefit from a formal statement with the relevant assumptions spelled out. In general, the uniqueness of the signature does not imply strict propriety of the kernel; while it's true that their augmentation removes invariances of time parameterization and translation, making the signature unique, this would only mean that the signature identifies the path and doesn't necessarily mean that the feature mean identifies the distribution. In my understanding, the proof would follow by assuming support on a compact path set (Props 3.1, 3.3 in Issa et al. 2023) and invoking standard results for the MMD.
- Claimed novelty: Signature kernel scores were already proposed by Issa et al. 2023 with theoretical justifications. This paper seems to be adapting the score to the weather forecasting application. Currently, Issa et al. 2023 is only referenced throughout the paper for various implementation choices, not in the main framing of the paper up front, which could be misleading.
- Clarity: While the paper explains paths, signatures, and the kernel trick in detail, it currently assumes significant background knowledge of the weather forecasting setting (e.g., using terms like IFS and AIFS without defining them). The experiments sections are quite dense with a lot of domain-specific details. Please see below for requested changes to improve clarity.
- Empirical validation: The experiments are partially convincing. The authors do convincingly demonstrate that it behaves differently from RMSE/CRPS and that it's sensitive to temporal structure. But there's no controlled demonstration, by way of e.g., simulations, that the score responds in expected ways under varying degrees of correlation. Also, spatial correlations are only partially (locally) tested due to patching. Practically, the method seems to suffer from numerical instability, requiring fixes like dimension-dependent rescaling.

**Requested Changes:**

Requested changes:
- Please see my suggestions above on making the theoretical claims more precise.
- The introduction should be modified to clearly isolate the contributions of this paper relative to prior work. In my understanding (please correct me if I misread), this paper is adapting the signature kernel score proposed by Issa et al. 2023 for the specific application of evaluating/training weather forecasting models. The phrasing in the current version may give the impression that it's being proposed for the first time, despite the reference in the introduction: "Via uniqueness guaranteed by the signature augmentations, the signature kernel score exhibits the desired fundamental properties of scoring rules, particularly being strictly proper, which ensures truthful predictions (Issa et al.,
2023)." I still do believe the paper provides meaningful contributions, by demonstrating the application of this scoring rule to (1) evaluating high-dimensional spatiotemporal forecasts and (2) to *training* a forecasting model (Issa et al. 2023 focus on evaluation), through prequential training and gradient evaluation with ensembles/PDE solver.
- The paper assumes quite a bit of domain knowledge in weather forecasting, requiring non-domain readers to look up terms or read between the lines. Please define all acronyms/abbreviations (e.g., IFS and AIFS). It would help to concretely state an example of one "forecast instance," maybe with a toy figure, and state the associated dimensionality (since "dimension" is overloaded here). Details of Weather Bench datasets and meaning of variables like z 500 or t 850 could use some descriptions in the Appendix. Some of the domain-specific details in the experiments sections could be moved to the Appendix, so that the main text can highlight key trends and the high-level setup -- for instance, it was difficult to appreciate at first that z 500 was a weather variable and that each weather variable was being forecasted separately, one at a time.
- The proposed eval/training pipeline has many components. Open-sourcing the code would facilitate understanding and adoption
- The "evaluation" section would be more convincing if the authors included the response of the various scores in a simulated setting, with known degrees of correlation. Currently, emphasis seems placed on how differently SIGK behaves from standard metrics, but it's not clear that it's necessarily correct.
- The RBF scale hyperparameter is set to 1 -- does this make sense given the scale of the augmented path elements?

Optional changes/questions:
- An algorithm pseudocode would be very helpful in putting together the various components of the pipeline, including the time augmentation, static RBF kernel computations, PDE solver, and the final signature kernel evaluation
- Could the authors comment on other forecasting applications with high-dimensional spatiotemporal fields that could benefit from using SIGK to train?

---

> ### Author Response · Authors · 2026-03-02
>
> Thank you very much for your patience during the revision process. We have submitted a revised version of the document to address your recommendations. In particular, we have included a proof of strict propriety in Appendix E and referenced with Theorem 1 in Section 2.4. We have more clearly established our contributions to the paper in the Introduction and have better cited the origins of the score. All acronyms and weather variables should now be defined, and we have added a link to the GitHub repository so the code is available to be run independently and all experiments can be recreated. Additionally, we have included a new section in the Appendix (G), where we use a simulated example to show how the signature kernel score responds to varied degrees of correlation compared to other scores.
>
> We hope you find that these additions strengthen the quality of the paper, and we welcome any further feedback.

---

> > ### Comment · Reviewer_z8ie · 2026-03-18
> >
> > Thank you for addressing my comments. I recommend acceptance and only have minor outstanding suggestions about the presentation of the proof in Sec E. The proof seems correct but currently reads more like a sketch. For showing compactness, it would be clearer to expand on how conditions for applying Arzela-Ascoli are met, especially equicontinuity. Currently, the proof just states: "paths being defined on the finite interval $[0, T]$ and [... being] piecewise linear with values in a bounded spatial domain [...] guarantees bounded total variation and equicontinuity."

---

> > > ### Author Response · Authors · 2026-03-20
> > >
> > > Thank you very much for the comment. We will formalise the proof following your suggestions and submit a revised version next week.

---

> > > ### Author Response · Authors · 2026-03-25
> > >
> > > Thank you for your patience. We have now submitted a revised version that expands the proof in section E, showing the compactness of the path space by explicitly verifying the three conditions of the Arzelà-Ascoli theorem.

---

### Review · Reviewer_dtzZ · 2026-02-16

**Summary Of Contributions:**

The paper applies the signature kernel method to weather data. It provides a high-level introduction to signatures and signature kernels, and uses the signature kernel score to train a weather-forecasting model.

(score updated after rebuttal, updates and discussion)

**Additional Comments:**

* p.2, section 2 is too long and contains a weird formulation “would”.

In this section, we would introduce a new diagnostic which is a strictly proper probabilistic scoring rule based on signatures of path that accounts for both temporal and spatial dependencies to adequately address the critical needs of probabilistic weather forecasting.

* p.4: Table 1 is hard to parse: tilted labels, linked names instead of name+proper citation
* p.9 “our uplifted” wrong wording
* p.9: The sentence below is paraphrased from the reference [3] (which paraphrases from [4]), and it is very unclear if one is not very familiar with related work. It seems like the actual meaning gets harder and to comprehend because the phrase “trivial ambient space” is very vague (I guess [3] refers to the linear kernel in [4, p.26]) and never defined and the reason why “lifting the underlying space would be a good learning strategy” ([4, p.3] mentions the ability to use featurizations of the ambient space that are known to work well.) is not explained futher.

sequential information often takes values in non-trivial ambient spaces, lifting the underlying space would be a good learning strategy regardless before considering the signature

* Figure 4 is of rather low quality and not described well. I would recommend properly describing the architecture mathematically and then turning Figure 4 into a block diagram referencing this description.

# References

[1]: Issa, Zacharia, et al. “Non-adversarial training of neural sdes with signature kernel scores.” Advances in Neural Information Processing Systems 36 (2023): 11102-11126.

[2]: Steinwart, Ingo, and Johanna F. Ziegel. “Strictly proper kernel scores and characteristic kernels on compact spaces.” Applied and Computational Harmonic Analysis 51 (2021): 510-542.

[3]: Salvi, Cristopher, et al. “The signature kernel is the solution of a goursat pde.” SIAM Journal on Mathematics of Data Science 3.3 (2021): 873-899.

[4]: Király, Franz J., and Harald Oberhauser. “Kernels for sequentially ordered data.” Journal of Machine Learning Research 20.31 (2019): 1-45.

**Audience:**

Yes

**Audience Explanation:**

The motivation given in the introduction seems highly interesting to me and motivates the investigation of unifying scoring rules:

> p.1: the area of weather forecasting is noticeably limited in the justification provided for the scoring rules in use. In current literature, very few papers are using scores suggested by meteorological agencies, particularly the European Centre for Medium-Range Weather Forecast (ECMWF), despite many using their IFS and AIFS models as references.

However, it is not actually addressed in the paper. The paper uses the kernel signature score as an objective to train a forecasting model, but does not investigate how training with these scores compares to the aforementioned variety of scores used in literature.

**Broader Impact Concerns:**

I have no concerns about a negative broader impact.

**Claims And Evidence:**

Yes

**Claims Explanation:**

The paper is written in a rather informal fashion. Important statements have no backing (reference/ simulation/ proof).

1. This statement lacks a justification: There are conditions on the domain of the kernel (compactness) and the kernel itself (characteristic) [2]. These properties are addressed in [1].

    > p.13: Critically, this is true of the signature kernel scoring rule, seen in terms of expectation above in Equation 4, as is standard for a general kernel score.

2. Missing reference.

    >p.8: There is evidence that the hierarchical dyadic windowing outperforms the global window.

3. The symbols Fi,j oi,j are never introduced.

    >p.9: φ(Fi,j,1:t, oi,j,1:t) = EX,X′∼Fi,j,1:t  [KSig(X, X ′)] − 2EX∼Fi,j,1:t  [KSig(X, oi,j,1:t)]

4. p.10: The kernel choice seems like the main culprit of using signature kernel scores. There is little insight into which kernel should be used in weather forecasts. The paper chooses the RBF kernel as a default, but does not investigate the effect of this choice.
5. The numerical instability issues discussed in Appendix C are interesting, but only described and not traced back to the actual root causes. They are thus hard to pick up and use, for example, for future work.
6. The choice to ignore sparsity, a property usually instrumental for spatiotemporal weather forecasting, seems not very convincing to me.
7. No backing.

    > p. 15: Unlike in evaluation, many standard weather models, such as the ones in Table 1, do not train on latitude weighted scores. As a result, we do not need to take latitude slices and can consider global paths or patches.

8. Not well explained. Why is this expected? Is the problem harder when forecasting for horizons? Should the path length not be the input to the kernel, and thus longer paths result in better performance?

    > As expected, smaller path lengths results in stronger performance, particularly notable in NRMSE and R2

9. The results of section 3 seem inconclusive at best, and I was not able to follow the reasoning of the authors that led them to conclude that the signature kernel score is a good score for weather forecast models.

**Requested Changes:**

1. The paper requires a complete reconsideration of the actual content and research question. So far, the paper presents itself as a hybrid between investigating the signature kernel score as a good score for training weather forecast models (section 3) and training a model based on the signature score to do weather forecasts (section 4). Neither is carried out in detail. In addition, both directions are rather simulation-heavy claims, and few simulations and benchmarks are conducted to back the claims of the paper.
    - Investigating the signature kernel score as a training objective as a good surrogate for the sophisticated, task-centric scoring  used, i.e., by ECMWF. This would include investigating the correlation of the signature kernel score to the task-centric scores and eventually training sota models with this score.
    - Building a signature-based weather forecasting model that exploits the structure and complies with the requirements of weather forecasting would also be interesting. This needs to include more detailed investigations and benchmarking against sota models.
2. In general, I think that this paper does not need to be a long submission. For example, cutting on the very introduction to signature kernels that contain no original contribution and focusing on the original contribution and findings backed by simulation experiment would significantly reduce the length and make the paper more accessible.

---

> ### Author Response · Authors · 2026-02-18
>
> Thank you for the review.
>
> The primary contribution of this work is to propose a diagnostic for long-timestep weather predictions, utilizing the path framing to score full forecast paths. In an attempt to make approaching this score more accessible and to help understand how it captures spatial and temporal correlations, we include the introductory section that leverages existing contributions on the derivation of the signature kernel, along with our novel contributions. However, we agree it is a lengthy section, and could be reduced.
>
> Model training using this score is a secondary component. We train with a lightweight model, which won’t compare to sota models, but demonstrates that it produces reasonable results. To benchmark this approach would have required further adaptation of existing methods, which could be novel on its own merit.
>
> We will definitely increase the reference coverage, and we appreciate the specific examples that require more justification. We missed the introduction to the symbol notation, and we will add that in. Additionally, we will fix the wording changes you highlighted.
>
> We are moving the implementation complications (kernel choice, dimensionality/scaling issues) up to the main text body and will expand on that discussion. We agree that uncertainty around the numerical instability challenges makes continuing investigation into this method challenging. Many hyperparameter choices were taken following Issa et al (2023), which result in greater stability, but don’t have much justification beyond heuristics.
>
> Your comment on sparsity is a valid point, and we will acknowledge that as potential for a future work. Sparsity within signature kernel metrics has only recently been explored by Redhead and Lee (2026).
>
> We will be submitting these updates in the next two weeks and look forward to further discussion.
>
> Benjamin R. Redhead, Thomas L. Lee, Peng Gu, Víctor Elvira, and Amos Storkey. Signature-kernel based evaluation metrics for robust probabilistic and tail-event forecasting, 2026. URL https://arxiv.org/abs/2602.10182.

---

> ### Author Response · Authors · 2026-03-02
>
> Thank you very much for your patience during the revision process. We have submitted a revised version of the document to address your recommendations. In particular, we have made the paper a normal length submission by moving the recommended content to the appendix. We have added references to unjustified statements, brought and expanded the numerical instability section into the main body (Section 4.3), and improved our wording in certain sections to improve understanding. We have clarified our primary contribution as introducing a diagnostic for long forecast horizons to the domain of weather forecasting.
>
> We hope you find that these additions strengthen the quality of the paper, and we welcome any further feedback.

---

> > ### Comment · Reviewer_dtzZ · 2026-03-04
> > **Reply to Rebuttal**
> >
> > Thank you for addressing much of the criticism brought up in my review.
> >
> > The paper, in my opinion, did improve substantially. There are still a couple of open issues I have listed below. If those and the concerns of reviewer __z8ie__ are appropriately addressed, I would be willing to recommend acceptance.
> >
> > # Open Issues
> > * In general, please perform a thorough spelling, formatting, and logic check. I was able to spot some inconsistencies, but I am sure I missed some.
> > * p.7.: FUXI is not explained and not spelled consistently.
> > * p.7: spelling
> > > In figure 1, the domain for ACC and MSE is
> >
> > * p.14.: Figure 3 is of too low quality. Please make sure to export it as a vectorized graphic or directly program it with a LaTeX package like pgfplots.
> > * I really appreciate the inclusion of simulation code. Please make sure that you accurately describe your environment (os you tested on, Python version, used packages), i.e. via a ‘pyproject.toml’, packaging or at least ‘pip freeze > requirements.txt’. Also, make sure to include the licence of your choice.
> > * p.12.: There is no proper limitation section.
> > * p.12.: I would move the reference to GitHub to page 1
> > > Specific details of implementation are found in the appendix ...
> >
> > * p.12.: $l >> d$, you probably want $l\gg d$
> > * p.22,23 - discriminative power:
> >   * It is not fully clear whether the signature score is beneficial or detrimental for training a forecasting model. In particular, there is currently no clear definition or metric for “discriminative power”. The analysis seems to rely largely on visual inspection of the deviation-score plot.
> >   * The “explosion” at positive deviation seems to already indicate the numerical issues, while the score is rather flat in regions of negative deviation, which the authors claim could lead to slow learning. If the analysis here is a good surrogate for training performance, comparing the different kernels inside SigKS would be natural.
> >   * In addition, the authors should justify the conjectures they make, by experiments or literature, c.f.
> >   > score are relatively flat near the actual optimum, which could result in slower, less optimal learning
> >
> > # Further questions
> >
> > * p.12.: Could you please investigate and elaborate on what this means in terms of computation and why the instabilities do or do not occur?
> > > We note that these instability issues are not present in the primary use of the signature kernel score as a diagnostic, where predicted paths remain closely tied to the distribution of observed weather.

---

> > > ### Author Response · Authors · 2026-03-05
> > >
> > > Thank you very much for the further review. We really appreciate your feedback, and we are committed to making the changes you have listed above. We will fix the quality issues, add the limitations section, and improve the discriminative power section.
> > >
> > > To answer your further question: we believe the numerical instability arises from two compounding factors, particularly the sharp loss landscape that can occur in training when predicted paths are significantly deviated from observations, and the accumulation of error in the Goursat PDE solver when path differences are large. These issues were evident with the linear kernel (and we will include these results in Appendix G), compared to the RBF, which alleviated them by largely dampening the magnitude of values. We will highlight this and link the result to the discriminative power section.
> > >
> > > We will make these changes, and once we incorporate the additional feedback from the other reviewers, we will promptly resubmit.

---

> > > ### Author Response · Authors · 2026-03-13
> > >
> > > Thank you for your continued patience. We have submitted a newly revised version addressing the open issues.
> > >
> > > We have replaced Figure 3 with a vectorised one constructed in TikZ. We have updated the Github with a license and added two environment setups for Windows or CentOS systems.
> > >
> > > We have added a limitations section and fixed the spelling issues mentioned, along with others.
> > >
> > > We have included an additional figure comparing static kernels in the discriminative power section. We have clarified the impact of the results from this section, particularly in relation to evaluation or training impact, and justified our conjectures.
> > >
> > > Finally, we have expanded upon the numerical instability section, addressing the sources of numerical instability in the method and how we prevented numerical instability in our final experiments.
> > >
> > > We believe these revisions address your concerns and have significantly strengthened the paper.

---

> > > > ### Comment · Reviewer_dtzZ · 2026-03-13
> > > > **Revision Received**
> > > >
> > > > Thank you for your revision. I will have a look at it during the next week.

---

> > > > ### Comment · Reviewer_dtzZ · 2026-03-25
> > > > **Final Recommendation Submitted**
> > > >
> > > > Thank you for the discussion and updates. I have no further questions and have submitted my final recommendation.

---

### Decision · Action_Editor_oimV · 2026-05-07

**Recommendation:** Accept as is

**Additional Comments:**

Noting that Figure j.7 prediction 3 doesn't render on my computer. Let's check that.

I also see some sort of broken cross page reference on page 21.

**Audience:**

Yes

**Audience Explanation:**

Forecasting is a core machine learning domain. Furthermore, with the explosion of interest in AI for science, machine learning for weather forecasting is very much in scope for TMLR. All reviewers agreed with this statement.

**Claims And Evidence:**

Yes

**Claims Explanation:**

In general, all three reviewers were reasonably positive by the end of the review process. I see somewhat more positive comments from reviewer zi8ie, who appreciated the empirical validations on weather forecasting, and by the end of the review process the theoretical support from the kernel. While there was no response from reviewer UqYp, their review expressed support for the technical and experimental claims of the manuscript. Finally, the negative claims from reviewer dtzZ were resolved through extensive discussion with the authors, in particular the ones about presentation of experiments (good job engaging to all involved!).

Overall, the experimental and theoretical claims are now reasonably well supported after the discussion with reviewers. Specifically:

a) there is now a full proof of strict propriety of the kernel and validation of the signature kernel score that seems to satisfy the reviewers.

b) there is now better discussion with related work from Issa et al, 2023.

c) the experimental results are now presented somewhat better.